# Aligning Task–Rank Preferences: Subspace Coverage and Anisotropy in LoRA Merging

## Abstract

Merging multiple Low-Rank Adaptation (LoRA) modules into a single model is a promising approach for constructing general-purpose systems, but it remains challenging because low-rank update directions introduced by LoRA adapters often span different subspaces and contribute unevenly across directions. When merged naively, such mismatches can weaken the directions most critical to certain task losses while overemphasizing relatively less important ones, ultimately reducing the model's ability to represent all tasks faithfully. We revisit this problem through two perspectives: subspace coverage, which captures how broadly LoRA directions cover diverse representational directions, and anisotropy, which reflects the imbalance of influence across those directions. We then propose TARA-Merging, short for *Task-Rank Anisotropy Alignment*. It explicitly incorporates task preferences by aligning the merging weights with a preference-weighted cross-entropy pseudo loss with preserving LoRA directions that encode task-relevant subspaces. This alignment ensures that the merged model maintains broad subspace coverage and accounts for anisotropy via direction-wise reweighting. Across eight vision and six NLI benchmarks, TARA-Merging consistently outperforms vanilla and LoRA-aware baselines, demonstrating strong robustness and generalization, and highlighting the importance of addressing both subspace coverage and anisotropy in LoRA merging.

## 1 Introduction

Low-Rank Adaptation (LoRA) (Hu et al., 2022) is widely used to adapt large foundation models to downstream tasks, including large language models and vision-language models such as CLIP (Radford et al., 2021). By introducing lightweight low-rank adapters, LoRA enables efficient fine-tuning with far fewer trainable parameters and reduced memory overhead, while mitigating the risk of overfitting when only limited data are available. This efficiency has encouraged training separate LoRA adapters on many different datasets, each capturing distinct aspects of vision or language understanding. Combining task-specific LoRA modules into a single network falls under the paradigm of *model merging* (Yang et al., 2024a), which provides an attractive alternative to costly multi-task training and enables the construction of general-purpose models that flexibly integrate knowledge across tasks and reflect different task preferences.

Recent research has explored various model merging. Early baselines such as task arithmetic (Ilharco et al., 2022; Ortiz-Jimenez et al., 2024; Jin et al., 2024) treat fine-tuned models as linear updates and combine them through simple interpolation. Ties-merging (Yadav et al., 2023) and DARE (Yu et al., 2024) reduces interference by dropping parameters, while AdaMerging (Yang et al., 2023) adapts merging coefficients using gradients derived from an entropy-based surrogate loss. Other lines of work include ensemble-based approaches (Kardan et al., 2021), permutation alignment techniques that resolve weight symmetries (Ainsworth et al., 2022; Entezari et al., 2021), and Pareto-aware optimization for preference-driven trade-offs (Chen & Kwok, 2024; Lin et al., 2024). More recently, LoRA-aware methods have been proposed: KnOTS (Stoica et al., 2025) aligns adapters via shared SVD bases, and LoRA-LEGO (Zhao et al., 2025) clusters adapters rank-wise to preserve modularity. While these approaches have advanced the field, they often overlook at least one of the two fundamental challenges in LoRA merging that we highlight in this work.

LoRA produces a low-rank update that we decompose into rank-wise components for analysis. We collectively term these directions as *LoRA directions*. With this setup, two aspects are essential for successful merging: *subspace coverage* and *anisotropy*. Subspace coverage captures how widely the LoRA directions collectively span the representational subspace, so preserving coverage ensures access to diverse task-relevant information. Anisotropy refers to directional sensitivity imbalance. Even with sufficient coverage, task losses can be unequally sensitive to different LoRA directions, which can distort trade-offs if ignored. Furthermore, when task preferences are specified, effective merging must preserve coverage while aligning merging weights with those preferences. Most merging methods address only one side of the problem. Task arithmetic and pruning reduce interference but erode subspace coverage and ignore direction-wise sensitivity. AdaMerging tunes global weights yet misses per-direction anisotropy. LoRA-aware methods like KnOTS and LoRA-LEGO improve structure but can shrink effective coverage and still lack sensitivity weighting. In short, they do not jointly preserve coverage and model anisotropy over LoRA directions.

Building on these observations, we propose TARA, short for *Task-Rank Anisotropy Alignment*. Our method introduces a structured merging framework that explicitly incorporates task preferences while addressing the two properties above. It preserves LoRA directions to maintain subspace coverage and accounts for anisotropy by direction-wise reweighting according to sensitivity. It then aligns the merging weights with a preference-weighted cross-entropy pseudo loss, yielding merged models that faithfully approximate fine-tuned models across tasks while improving robustness and generalization. We validate our approach across eight vision datasets and six natural language inference tasks, where TARA consistently outperforms vanilla baselines and recent LoRA-aware methods in joint-task evaluation and transfer to unseen tasks.

Our main contributions are summarized as follows:

- Identification of **subspace coverage** and **anisotropy** as two key properties for analyzing and guiding LoRA merging, highlighting their roles in subspace preservation and direction-wise sensitivity.
- Introduction of **TARA**, a framework that aligns both task-level and rank-level preferences, ensuring coverage of LoRA subspaces while mitigating anisotropy-induced imbalance.
- Extensive empirical validation on vision and language benchmarks, demonstrating state-of-the-art performance and strong robustness in various experimental settings.

## 2 ANALYZING SUBSPACE COVERAGE AND ANISOTROPY IN LORA MERGING

### 2.1 PROBLEM SETTING AND NOTATION

We formalize the problem of *LoRA merging*, a special case of model merging where the goal is to combine multiple LoRA fine-tuned adapters into a single model that balances performance across tasks according to user-specified preferences.

**Base model and LoRA adapters.** Let $W_0 \in \mathbb{R}^{d \times m}$ be the frozen pretrained weight matrix. For each task $i \in \{1, \ldots, N\}$, we attach a LoRA adapter of rank $r_i$, parameterized as

$$\Delta W_i = B_i A_i^\top, \qquad B_i \in \mathbb{R}^{d \times r_i}, \ A_i \in \mathbb{R}^{m \times r_i}. \tag{1}$$

Thus, $\mathrm{rank}(\Delta W_i) \leq r_i \ll \min(d, m)$. We further decompose each as $\Delta W_i = \sum_{j=1}^{r_i} b_{ij} a_{ij}^\top$, where $b_{ij}$ and $a_{ij}$ denote the $j$-th columns of $B_i$ and $A_i$. Each outer product $b_{ij} a_{ij}^\top$ is referred to as a *rank-1 LoRA direction*.

**User preferences.** We assume that the relative importance of tasks is specified by a vector $\rho = [\rho_1, \ldots, \rho_N]^\top \in \Delta_{N-1}$, where $\Delta_{N-1}$ denotes the probability simplex. A larger $\rho_i$ indicates higher priority for task $i$ in the merged model.

**Merging as Multi-Objective Optimization.** Without fixing preferences, LoRA merging can be cast as a multi-objective optimization (MOO) problem (Miettinen, 1999) that seeks a set of solutions. We minimize the objective vector $f(W) = [f_1(W), \ldots, f_N(W)]^\top$ over merged models $W$ obtained by linearly combining available LoRA adapters on top of a fixed base $W_0$. A solution $W^a$ *dominates* $W^b$ if $f_k(W^a) \leq f_k(W^b)$ for all $k \in \{1, \ldots, N\}$ and $f_j(W^a) < f_j(W^b)$ for some $j$. A solution is *Pareto optimal* if no other feasible solution dominates it. The *Pareto front* is the collection of Pareto-optimal solutions.

## 2.2 Preserving LoRA Subspace Coverage

We quantify how much task-specific representational capacity is retained when LoRA adapters are merged. We refer to this as *subspace coverage*: it measures how broadly the task-specific LoRA directions span the parameter space and how well a merge preserves that span. We measure subspace coverage using the entropy-based *effective rank* (erank) (Roy & Vetterli, 2007). Let $X \in \mathbb{R}^{d \times m}$ have singular values $\{\sigma_k\}_{k=1}^{r_X}$ where $r_X = \text{rank}(X)$, and define $p_k = \sigma_k^2 / \sum_{j=1}^{r_X} \sigma_j^2$. Then

$$\text{erank}(X) = \exp\Big(-\sum_{k=1}^{r_X} p_k \log p_k\Big).$$

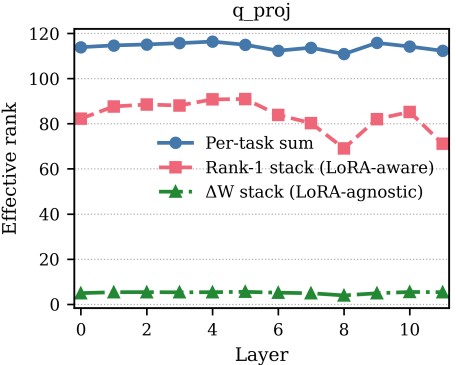

Figure 1: **Effective rank** across layers for an attention projection layer (e.g., query). The gap between $\Delta X$ *stack* and *Rank-1 stack* reflects merge-induced collapse.

$\text{erank}(X)$ equals $r_X$ for a flat spectrum and decreases as energy concentrates, providing a basis-invariant measure of effective dimensionality.

We compare three quantities using effective rank, with all stacks formed by vectorizing matrices and placing one vector per row. First, *per-task sum* computes $\text{erank}$ for each task $i$ on $X_i = [\text{vec}(b_{i1} a_{i1}^\top), \ldots, \text{vec}(b_{ir_i} a_{ir_i}^\top)]^\top$ and then sums $\sum_{i=1}^N \text{erank}(X_i)$. Second, the $\Delta X$ stack (*LoRA-agnostic*) uses $X_{\text{agnostic}} = [\text{vec}(\Delta W_1), \ldots, \text{vec}(\Delta W_N)]^\top$ and computes a single $\text{erank}(X_{\text{agnostic}})$. Third, the Rank-1 stack (*LoRA-aware*) aggregates all rank-1 factors across tasks into $X_{\text{aware}} = [\text{vec}(b_{11} a_{11}^\top), \ldots, \text{vec}(b_{Nr_N} a_{Nr_N}^\top)]^\top$ and computes $\text{erank}(X_{\text{aware}})$. Here we refer to the Rank-1 stack as *LoRA-aware* because it preserves the adapter factorization and treats each rank-1 direction $b_{ij} a_{ij}^\top$ as a separate basis element, whereas the $\Delta W$ stack is *LoRA-agnostic* since it collapses each adapter into a single update $\Delta W_i$ and ignores the internal rank-1 structure.

As shown in Fig. 1, the *LoRA-aware* stack retains about 70% of the *per-task sum*, indicating that most task-specific directions remain approximately independent after redundancy removal and thus cross-task alignment is weak. This agrees with prior work showing that LoRA fine-tuned models align less across tasks than full-rank fine-tuning (Stoica et al., 2025). The gap between the *LoRA-aware* and *LoRA-agnostic* stacks captures subspace collapse under interpolation-based merging (Ilharco et al., 2022; Yadav et al., 2023; Yang et al., 2023), because unlike $X_{\text{aware}}$ the interpolated updates in $X_{\text{agnostic}}$ can interfere destructively across tasks and lower the effective rank. Formal definitions and additional results appear in Appendix D.

## 2.3 Anisotropy of LoRA Directions

Let $W$ be the current weights and $\{S_k\}_{k=1}^K$ a set of *LoRA directions*, where a LoRA adapter uses factor matrices $B \in \mathbb{R}^{d \times r}$ and $A \in \mathbb{R}^{m \times r}$ (for simplicity, the same rank $r$ for all adapters), with columns $b_k$ and $a_k$, and we set $S_k = b_k a_k^\top$ (so $K = r$ for a single adapter, or the sum of ranks if multiple adapters are stacked). Consider a small update restricted to this span with *direction-selection coefficients* $\phi = (\phi_1, \ldots, \phi_K)^\top \in \mathbb{R}^K$ (distinct from the task-preference vector $\rho$), so that $\Delta W = \sum_{k=1}^K \phi_k S_k$. Using the Frobenius inner product, the first-order change of each task loss is

$$\Delta f_i \approx \big\langle \nabla f_i(W), \Delta W \big\rangle_F = \sum_{k=1}^K \phi_k \big\langle \nabla f_i(W), S_k \big\rangle_F. \qquad (2)$$

Stacking $\Delta f = [\Delta f_1, \ldots, \Delta f_N]^\top$ yields the linearization

$$\Delta f \approx J \phi, \qquad J_{i,k} = \big\langle \nabla f_i(W), S_k \big\rangle_F, \qquad (3)$$

so $J(W)$ is the task-loss Jacobian restricted to $\text{span}\{S_k\}$. Directional imbalance is captured by the singular values of $J$, which give the following bound from coefficients to loss changes.

**Proposition 1** (Anisotropy Bounds). *Let $\sigma_{\max}(J)$ and $\sigma_{\min}(J)$ denote the largest and smallest singular values of $J$. Then, for any coefficient vector $\phi$,*

$$\sigma_{\min}(J) \|\phi\|_2 \leq \|J\phi\|_2 = \|\Delta f\|_2 \leq \sigma_{\max}(J) \|\phi\|_2. \qquad (4)$$

*When $\kappa(\boldsymbol{J}) = \sigma_{\max}(\boldsymbol{J})/\sigma_{\min}(\boldsymbol{J})$ is large, the map $\phi \mapsto \Delta\boldsymbol{f}$ is* anisotropic*, and equal-norm LoRA-direction updates need not yield proportional task-loss changes.*

**Empirical evidence of anisotropy.** To illustrate the qualitative effect anticipated by Proposition 1, we measure singular-value spectra and condition numbers of $\boldsymbol{J}$ across layers and modules. We observe a pronounced spectral spread and large condition numbers, indicating that a few directions dominate the task-loss response while many are weak. Full plots are deferred to Appendix E; see the condition-number traces in Figs. 6 and 7 and the scree curves in Fig. 8.

**Directional-sensitivity misalignment across preferences.** With the scalarized gradient $g(\boldsymbol{\rho}; \boldsymbol{W}) = \sum_i \rho_i \nabla f_i(\boldsymbol{W})$, define direction-wise *loss sensitivities*

$$h_k(\boldsymbol{\rho}; \boldsymbol{W}) = \langle g(\boldsymbol{\rho}; \boldsymbol{W}), \boldsymbol{S}_k \rangle_F, \qquad \boldsymbol{h}(\boldsymbol{\rho}; \boldsymbol{W}) = (h_1, \ldots, h_K)^\top. \tag{5}$$

When the preference $\boldsymbol{\rho}$ changes, the pattern of $\boldsymbol{h}(\boldsymbol{\rho}; \boldsymbol{W})$ also changes as well. We quantify *Directional Sensitivity Misalignment* by

$$\xi(\boldsymbol{\rho}_1, \boldsymbol{\rho}_2; \boldsymbol{W}) = 1 - \frac{|\langle \boldsymbol{h}(\boldsymbol{\rho}_1; \boldsymbol{W}), \boldsymbol{h}(\boldsymbol{\rho}_2; \boldsymbol{W}) \rangle|}{\|\boldsymbol{h}(\boldsymbol{\rho}_1; \boldsymbol{W})\|_2 \|\boldsymbol{h}(\boldsymbol{\rho}_2; \boldsymbol{W})\|_2}, \tag{6}$$

so that $\xi \in [0, 1]$. Here $\langle \cdot, \cdot \rangle$ denotes the dot product. Smaller $\xi$ indicates weaker $\boldsymbol{\rho}$-dependence of the sensitivity profiles, whereas larger $\xi$ indicates stronger $\boldsymbol{\rho}$-dependence. When measuring anisotropy, a $180°$ flip lies in the same one-dimensional span. We therefore treat sign flips as equivalent and focus on alignment along the direction axis.

We quantify how loss-sensitive directions within the LoRA span change when the task preference switches from $\boldsymbol{\rho}_1$ (uniform over tasks) to $\boldsymbol{\rho}_2$ (one-hot on a single task). For each preference, we form a sensitivity profile over LoRA directions by projecting the scalarized gradient $g(\boldsymbol{\rho}; W) = \sum_i \rho_i \nabla f_i(\mathbf{X})$ onto the basis $\{\boldsymbol{S}_k\}$. Throughout this analysis, we set $W$ to the Task Arithmetic merge $\boldsymbol{W}_{\mathrm{merge}} = \boldsymbol{W}_0 + \lambda \sum_{i=1}^N \Delta\boldsymbol{W}_i$ with $\lambda = 0.3$ following (Ilharco et al., 2022). We then summarize the distance between the two preference with a misalignment index $\xi(\boldsymbol{\rho}_1, \boldsymbol{\rho}_2) \in [0, 1]$. Figure 2 shows substantial misalignment across layers and modules, indicating that directional sensitivities within the LoRA span are highly preference-dependent and motivating our subsequent directional alignment.

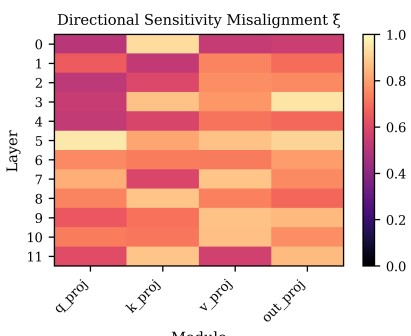

Figure 2: **Directional-sensitivity misalignment $\xi(\boldsymbol{\rho}_1, \boldsymbol{\rho}_2)$.** Larger $\xi$ indicates stronger change in loss-sensitive directions when switching preferences.

## 3 METHOD

Based on the analysis, we propose TARA, *Task-Rank Anisotropy Alignment*, which addresses both *subspace coverage* and *anisotropy* in LoRA merging. TARA weights LoRA directions with an entropy-minimization term, as in AdaMerging (Yang et al., 2023). We present two variants that specify how the directions are constructed and how the direction-selection weights are assigned.

**Variant A: Per-rank LoRA direction selection.** This variant reweights individual rank-1 factors and focuses on anisotropy control within each adapter. Let $\phi_{ij} \in \mathbb{R}$ be a *direction-selection weight* for factor $(i, j)$ with $i \in \{1, \ldots, N\}$ and $j \in \{1, \ldots, r_i\}$. The merged weight is

$$\boldsymbol{W}_A(\phi) = \boldsymbol{W}_0 + \sum_{i=1}^N \sum_{j=1}^{r_i} \phi_{ij} \, \boldsymbol{b}_{ij} \, \boldsymbol{a}_{ij}^\top. \tag{7}$$

Unless otherwise noted, we allow signed weights and learn their scale directly.

**Variant B: Shared singular-direction selection.** This variant builds a shared orthonormal basis across tasks to preserve subspace coverage, reduce cross-task interference, and control anisotropy via weights. Form a matrix by *horizontally* concatenating adapters

$$\boldsymbol{X} = \begin{bmatrix} \Delta\boldsymbol{W}_1, \ldots, \Delta\boldsymbol{W}_N \end{bmatrix} \in \mathbb{R}^{d\times(mN)},$$

and compute the SVD: $\boldsymbol{X} = \boldsymbol{U}\boldsymbol{\Sigma}\boldsymbol{V}^\top$. Select the top $R$ singular values and their associated singular vectors, yielding

$$\boldsymbol{U} = [\boldsymbol{u}_1, \ldots, \boldsymbol{u}_R] \in \mathbb{R}^{d\times R}, \quad \boldsymbol{\Sigma} = \mathrm{diag}(\sigma_1, \ldots, \sigma_R), \quad \boldsymbol{V} = [\boldsymbol{v}_1, \ldots, \boldsymbol{v}_R] \in \mathbb{R}^{(mN)\times R}.$$

Partition each $\boldsymbol{v}_k$ into $\boldsymbol{v}_k = [\,\boldsymbol{v}_{k1}^\top, \ldots, \boldsymbol{v}_{kN}^\top\,]^\top$ with $\boldsymbol{v}_{ki} \in \mathbb{R}^m$ and define

$$\boldsymbol{S}_{ik} = \boldsymbol{u}_k\,\boldsymbol{v}_{ki}^\top \in \mathbb{R}^{d\times m} \qquad (i=1,\ldots,N;\; k=1,\ldots,R). \tag{8}$$

Assign weights $\phi_{ik} \in \mathbb{R}$ to these directions and obtain

$$\boldsymbol{W}_B(\boldsymbol{\phi}) = \boldsymbol{W}_0 + \sum_{i=1}^N \sum_{k=1}^R \phi_{ik}\,\sigma_k\,\boldsymbol{S}_{ik} = \boldsymbol{W}_0 + \sum_{i=1}^N \sum_{k=1}^R \phi_{ik}\,\sigma_k\,\boldsymbol{u}_k\,\boldsymbol{v}_{ki}^\top. \tag{9}$$

Together, the shared orthonormal left directions $\{\boldsymbol{u}_k\}_{k=1}^R$ and the per-task partitions of the right singular vectors preserve subspace coverage. The left basis $\{\boldsymbol{u}_k\}$ retains the top-$R$ column space of $\boldsymbol{X}$, and $\boldsymbol{v}_{ki} \in \mathbb{R}^m$ yields *rank-1 components* $\boldsymbol{S}_{ik} = \boldsymbol{u}_k\boldsymbol{v}_{ki}^\top$. Direction weights then adjust these orthonormal rank-1 components preserving subspace coverage.

Following Lin et al. (2024), we optimize the direction-selection weights $\boldsymbol{\phi}$ under a preference vector $\boldsymbol{\rho}$. Let $\boldsymbol{W}(\boldsymbol{\phi})$ denote either $\boldsymbol{W}_A(\boldsymbol{\phi})$ from eq. (7) or $\boldsymbol{W}_B(\boldsymbol{\phi})$ from eq. (9).

**Smooth Tchebycheff Scalarization.** With anchors $z_i = \min_{\boldsymbol{W}} f_i(\boldsymbol{W})$, define

$$\Psi(\boldsymbol{\phi}, \boldsymbol{\rho}) = \alpha \log\left(\sum_{i=1}^N \exp\left(\frac{\rho_i\,|f_i(\boldsymbol{W}(\boldsymbol{\phi})) - z_i|}{\alpha}\right)\right), \qquad \alpha > 0. \tag{10}$$

Here $z_i$ acts as a task-wise anchor that normalizes scale differences across tasks. The parameter $\alpha$ controls smoothing: *smaller* $\alpha$ concentrates the objective on the worst-performing task, whereas *larger* $\alpha$ yields a smoother aggregation that spreads focus more evenly across tasks. During merging, we use predictive entropy following AdaMerging (Yang et al., 2023) since $f_i$ requires labels. For $z_i$, we use the entropy loss obtained when only task $i$'s adapter is applied.

## 4 EXPERIMENTS

**Experimental Setup.** For vision experiments, we follow Stoica et al. (2025) and use CLIP (Radford et al., 2021) with a ViT-B/32 backbone (Dosovitskiy et al., 2020) pre-trained on ImageNet-21k (Deng et al., 2009). Models are fine-tuned with rank-16 LoRA unless noted otherwise and evaluated on eight image classification benchmarks. We also test our method on six classification datasets with the LLaMA-3 8B model (Dubey et al., 2024). Dataset descriptions are in Appendix C.1, and LoRA fine-tuning details in Appendix C.2.

**Baselines.** We denote the pretrained weights by $\boldsymbol{W}_0$, the task vector of the $i$-th task by $\Delta\boldsymbol{W}_i$, and the merged weights by $\boldsymbol{W}_{\mathrm{merge}}$. For LoRA task vectors, we write $\Delta\boldsymbol{W}_i = \boldsymbol{B}_i\boldsymbol{A}_i^\top$. The scaling coefficient for merging is denoted by $\lambda$. Our evaluation spans two distinct groups of baselines. *Vanilla merging* comprises (i) **RegMean** (Jin et al., 2022) aligns the merged model with fine-tuned models by solving a layer-wise least-squares problem using small calibration sets from each task, thereby matching the activations of the merged model to those of the task-specific models. (ii) **Task Arithmetic (TA)** (Ilharco et al., 2022) adds a scaled sum of task vectors to the pretrained weights, $\boldsymbol{W}_{\mathrm{merge}} = \boldsymbol{W}_0 + \lambda \sum_{i=1}^N \Delta\boldsymbol{W}_i$. (iii) **TIES** (Yadav et al., 2023) prunes small-magnitude parameters, resolves sign conflicts, and then averages only parameters with consistent signs with scaling coefficient $\lambda$. (iv) **DARE-TIES** (Yu et al., 2024) sparsifies each parameter with Bernoulli probability $p$ and rescales the retained parameters by $1/(1-p)$ to preserve the expectation. (v) **AdaMerging** (Yang et al., 2023) learns merging coefficients by minimizing an output-entropy surrogate in the

Table 1: Per-task accuracy on eight image-classification benchmarks. We merge eight ViT-B/32 checkpoints, each fine-tuned with LoRA. The upper panel shows the per-task absolute accuracy of the fine-tuned baselines. The lower panel shows the accuracy of the merged models, normalized by their corresponding fine-tuned baseline (%).

| Method | Dataset | | | | | | | | |
|---|---|---|---|---|---|---|---|---|---|
| | Cars | DTD | EuroSAT | GTSRB | MNIST | RESISC45 | SUN397 | SVHN | Avg |
| | *Per-task absolute accuracies (%)* | | | | | | | | |
| Finetuned | 74.0 | 58.3 | 99.0 | 92.7 | 99.3 | 88.4 | 64.5 | 96.2 | 84.1 |
| | *Per-task accuracies of merged models, normalized to finetuned (%)* | | | | | | | | |
| *Vanilla Merging-Gradient Free* | | | | | | | | | |
| RegMean | 80.2 | 71.3 | 37.9 | 47.3 | 43.1 | 70.5 | 93.9 | 43.0 | 60.9 |
| TA | 82.1 | 74.3 | 48.7 | 41.8 | 53.4 | 71.5 | 96.6 | 42.0 | 63.8 |
| TIES | 81.0 | 72.5 | 53.8 | 37.4 | 69.0 | 65.3 | 94.8 | 45.3 | 64.9 |
| DARE-TIES | 81.6 | 74.5 | 51.0 | 37.2 | 59.2 | 66.6 | 96.0 | 38.8 | 63.1 |
| *Vanilla Merging-Gradient Based* | | | | | | | | | |
| AdaMerging (250 Iters) | 82.5 | 75.2 | 66.2 | 37.3 | 73.9 | 72.0 | 97.0 | 61.9 | 70.7 |
| AdaMerging (500 Iters) | 79.5 | 73.5 | 70.9 | 39.7 | 63.0 | 69.0 | 97.8 | 66.6 | 70.0 |
| *LoRA-aware Merging-Gradient Free* | | | | | | | | | |
| SVD | 81.8 | 73.5 | 47.7 | 38.3 | 52.8 | 70.2 | 96.4 | 40.4 | 62.6 |
| Linear | 81.4 | 74.5 | 49.5 | 43.0 | 55.3 | 70.3 | 96.8 | 39.0 | 63.7 |
| KnOTS-TIES | 82.7 | 73.7 | 49.3 | **48.9** | 68.9 | 70.9 | 95.5 | 53.8 | 68.0 |
| KnOTS-DARE-TIES | 81.8 | 75.9 | 50.7 | 40.3 | 53.2 | 70.2 | 97.9 | 41.0 | 63.9 |
| LoRA-LEGO | 81.1 | 73.0 | 54.4 | 40.3 | 48.6 | 71.5 | 97.3 | 37.1 | 62.9 |
| *LoRA-aware Merging-Gradient Based* | | | | | | | | | |
| TARA-Variant A (250 Iters) | 84.5 | 76.2 | 68.9 | 39.4 | 82.2 | 72.8 | 97.5 | 70.0 | 73.9 |
| TARA-Variant A (500 Iters) | 82.2 | 76.0 | 74.9 | 43.5 | 76.3 | 70.2 | 98.0 | **70.8** | 74.0 |
| TARA-Variant B (250 Iters) | **86.2** | **78.4** | 76.8 | 42.9 | **82.7** | **75.4** | **98.6** | 69.7 | **76.3** |
| TARA-Variant B (500 Iters) | 85.4 | 77.1 | **79.2** | 46.5 | 73.7 | 73.3 | **98.6** | 50.8 | 73.1 |

spirit of test-time adaptation (Wang et al., 2020). *LoRA-aware merging* exploits low-rank adapter structure: (vi) **SVD** aggregates LoRA task vectors $\Delta W_{\text{merge}} = \lambda \sum_i B_i A_i^\top$ and applies truncated SVD $\Delta W_{\text{merge}} \approx U_r \Sigma_r V_r^\top$ (Tang et al., 2025) to recover rank of original LoRAs (vii) **Linear** performs TA on $\{B_i\}$ and $\{A_i\}$ rather than whole task vector. (viii) KnOTS (Stoica et al., 2025) computes an SVD on concatenated task vectors $[\Delta W_1; \ldots; \Delta W_N] = U \Sigma V^\top$ and merge partitions of $V$ associated with each task vector. TIES or DARE-TIES can be applied to the $V_i$, yielding the variants **KnOTS-TIES** and **KnOTS-DARE-TIES**. (ix) **LoRA-LEGO** (Zhao et al., 2025) decomposes each adapter into minimal semantic units, clusters them rank-wise, and assembles a new LoRA from the cluster centroids. We present a faithful reimplementation of LoRA-LEGO. Detailed explanations and implementation specifics for all baselines can be found in Appendix C.3.

**Implementation Details.** For our method, we adopt AdamW (Loshchilov & Hutter, 2019) with a learning rate of 0.001. The direction-selection weights $\phi$ are initialized to 0.4 and optimized with a batch size of 16 for 500 iterations similar to Yang et al. (2023) to control training cost. For experiments, we set $\alpha = 1$ in eq. (10).

**Metrics.** Following prior work (Ilharco et al., 2022; Yadav et al., 2023; Stoica et al., 2025), we report the absolute accuracy of individually fine-tuned models on each dataset, and compare different merging methods using a normalized accuracy metric. The normalized accuracy is defined as $\frac{\text{Accuracy of merged model on task-}i}{\text{Accuracy of fine-tuned model on task-}i}$. This metric indicates how closely a merged model approaches the performance of the corresponding fine-tuned model for each task.

## 4.1 EXPERIMENTAL RESULTS

**Per-Task Evaluation across Vision Tasks.** In this experiments, we adopt the conventional per-task evaluation protocol (Ilharco et al., 2022; Yadav et al., 2023; Stoica et al., 2025), where the goal is to test how well different merging strategies preserve task-specific performance. Concretely, a collection of models fine-tuned independently on separate datasets are merged into a unified model, and its performance is measured on each dataset separately, relying solely on the dataset's own samples and labels. Table 1 reports normalized per-task accuracies and their averages across eight vision tasks with CLIP, where all models are fine-tuned with LoRA of rank 16. We categorize the baselines into two groups: vanilla merging methods and LoRA-aware methods that explicitly incorporate the LoRA structure. LoRA-aware methods generally outperform their vanilla counterparts. For instance, KnOTS-TIES surpasses TIES, and KnOTS-DARE-TIES improves upon DARE-TIES. KnOTS-based variants highlight the benefits of accounting for LoRA structure. LoRA-LEGO does

Table 2: Per-task accuracy on six NLI benchmarks. We merge six LLaMA-3 8B checkpoints, each fine-tuned with LoRA. The upper panel shows the absolute per-task accuracy of the fine-tuned baselines. The lower panel shows the merged models' accuracy, normalized by each task's corresponding fine-tuned baseline (%).

| Method | Dataset | | | | | | |
|---|---|---|---|---|---|---|---|
| | MNLI | QNLI | SNLI | RTE | SICK | SCITAIL | Avg |
| | *Per-task absolute accuracies (%)* | | | | | | |
| Finetuned | 90.8 | 95.3 | 92.1 | 84.8 | 91.3 | 87.3 | 90.2 |
| | *Per-task accuracies of merged models, normalized to finetuned (%)* | | | | | | |
| ***Vanilla Merging-Gradient Free*** | | | | | | | |
| TA | 67.3 | 87.3 | 41.8 | 95.7 | 77.9 | 76.9 | 74.6 |
| TIES | 61.3 | 91.5 | 38.2 | 85.5 | 76.8 | 70.5 | 70.6 |
| DARE-TIES | 42.0 | 72.5 | 44.1 | 77.8 | 76.9 | 86.8 | 66.7 |
| ***Vanilla Merging-Gradient Based*** | | | | | | | |
| AdaMerging (250 Iters) | 55.3 | 91.6 | 41.3 | 95.7 | 92.2 | 86.5 | 77.1 |
| AdaMerging (500 Iters) | 47.5 | 92.9 | 41.3 | 102.6 | 93.8 | 94.2 | 78.7 |
| ***LoRA-aware Merging-Gradient Free*** | | | | | | | |
| SVD | 67.7 | 87.2 | 41.7 | 95.7 | 77.6 | 76.6 | 74.4 |
| Linear | 63.1 | 86.2 | 39.5 | 90.6 | 77.5 | 74.5 | 71.9 |
| KnOTS-TIES | 41.1 | 83.4 | **56.6** | 87.18 | 87.9 | 94.8 | 75.2 |
| KnOTS-DARE-TIES | **76.4** | 88.2 | 39.9 | 99.2 | 79.6 | 75.6 | 76.5 |
| LoRA-LEGO | 62.5 | 84.8 | 41.4 | 100.0 | 87.0 | 83.7 | 76.6 |
| ***LoRA-aware Merging-Gradient Based*** | | | | | | | |
| TARA-Variant A (250 Iters) | 57.2 | 91.4 | 41.3 | 96.6 | 92.7 | 86.3 | 77.6 |
| TARA-Variant A (500 Iters) | 51.7 | 92.6 | 41.4 | 102.6 | 95.3 | 94.4 | 79.7 |
| TARA-Variant B (250 Iters) | 51.7 | 91.7 | 43.1 | 100.0 | 95.2 | 92.7 | 79.1 |
| TARA-Variant B (500 Iters) | 46.8 | **94.1** | 41.4 | **103.4** | **98.1** | **97.8** | **80.3** |

not provide significant advantages over vanilla methods, suggesting that merely preserving LoRA rank-1 directions is insufficient to retain task-specific information during merging. By comparison, TARA consistently outperforms both vanilla and LoRA-aware baselines. Variant A shows steady gains with additional training iterations, leading to higher average accuracy. Variant B achieves the best overall trade-off, delivering the highest scores average of 76.3% with only 250 iterations. These results demonstrate that exploiting LoRA structure is essential for stable merging, and that TARA further effectively capture task-sensitive subspaces. Experiments in Tab. 1 use the checkpoints released by KnOTS (Stoica et al., 2025). We further confirm the results on independently trained checkpoints, as shown in Tab. 5.

**Per-Task Evaluation on 6 NLI Tasks with LLMs.** In Tab. 2, we report per-task normalized accuracies of merged models on six NLI benchmarks. For this setting, we apply six LoRA adapters with rank 16 to LLaMA-3-8B (Dubey et al., 2024). Each benchmark follows the natural language inference protocol, where a hypothesis is classified against a given premise into entailment, contradiction, or neutral. Detailed descriptions of each dataset are provided in Appendix C.1. The NLI results with LLMs reveal trends consistent with those observed in vision tasks. Vanilla merging baselines, with the exception of AdaMerging, show limited ability to preserve task-specific information in large-scale models, while LoRA-aware methods still provide only moderate improvements. In contrast, TARA delivers significant and consistent gains. Variant B (500 Iters) achieving the best average normalized accuracy of 80.3%, highlighting the effectiveness of TARA on language tasks.

**Joint Task Evaluation of General Models.** The joint-task evaluation protocol, originally introduced by Stoica et al. (2025), differs from the per-task setup by evaluating merged models over the union of inputs and labels from all eight vision benchmarks. This setting is more suitable for assessing whether a merged model can serve as a general-purpose model across multiple target datasets. In constructing the joint task, labels from all benchmarks are pooled together and duplicates are removed, resulting in a unified label space. A detailed description of this setting is provided in Appendix C.4. Table 3 presents joint-task evaluation results in terms of top-1/3/5 accuracies. Ensemble (Kardan et al., 2021) and vanilla

Table 3: **Joint-Task Evaluation Results.** All models share a ViT-B/32 backbone, are fine-tuned with LoRA on separate datasets, and are then merged for joint-task evaluation.

| Method | Hits@1 | Hits@3 | Hits@5 |
|---|---|---|---|
| Ensemble | 40.7 | 63.1 | 72.6 |
| TA | 43.5 | 65.2 | 74.0 |
| TIES | 43.6 | 65.3 | 73.9 |
| DARE-TIES | 44.0 | 66.4 | 75.1 |
| AdaMerging (250 Iters) | 46.7 | 68.9 | 78.5 |
| AdaMerging (500 Iters) | 48.1 | 73.2 | 83.0 |
| KnOTS-TIES | 46.8 | 68.1 | 76.3 |
| KnOTS-DARE-TIES | 45.2 | 66.9 | 75.3 |
| LoRA-LEGO | 43.1 | 65.0 | 73.9 |
| TARA-Variant A (250 Iters) | 48.7 | 70.5 | 79.7 |
| TARA-Variant A (500 Iters) | **51.1** | **75.9** | 84.9 |
| TARA-Variant B (250 Iters) | 50.6 | 73.5 | 82.6 |
| TARA-Variant B (500 Iters) | 49.3 | 74.9 | **85.1** |

Table 4: Generalization results on two unseen tasks when merging ViT-B/32 models on six tasks.

| | Seen Tasks | | | | | | | Unseen Tasks | | | All Tasks |
|---|---|---|---|---|---|---|---|---|---|---|---|
| Method | Cars | DTD | GTSRB | RESISC45 | SUN397 | SVHN | Avg Acc | EuroSAT | MNIST | Avg Acc | Avg Acc |
| | *Per-task accuracies of merged models, normalized to finetuned (%)* | | | | | | | | | | |
| TA | 82.0 | 74.7 | 42.2 | 71.1 | 96.8 | 42.2 | 68.2 | 37.5 | 48.2 | 42.9 | 61.8 |
| TIES | 81.5 | 73.2 | 40.3 | 64.4 | 95.4 | 45.7 | 66.7 | 37.2 | 61.8 | 49.5 | 62.4 |
| KnOTS-TIES | 82.7 | 74.4 | 48.5 | 72.5 | 95.3 | 50.2 | 70.6 | 33.2 | 50.5 | 41.8 | 63.4 |
| KnOTS-DARE-TIES | 81.7 | 72.5 | **49.4** | 71.9 | 94.6 | 52.5 | 70.4 | 31.6 | 51.2 | 41.4 | 63.2 |
| LoRA-LEGO | 82.2 | 73.5 | 46.9 | 71.2 | 96.7 | 40.8 | 68.5 | 35.5 | 44.7 | 40.1 | 61.4 |
| AdaMerging (250 Iters) | 83.0 | 75.0 | 38.6 | 71.6 | 97.3 | 59.6 | 70.9 | 41.5 | 61.2 | 51.4 | 66.0 |
| AdaMerging (500 Iters) | 79.7 | 73.4 | 37.7 | 69.8 | 97.9 | 67.4 | 71.0 | 48.7 | 58.7 | 53.7 | 66.7 |
| TARA-Variant A (250 Iters) | 84.9 | **75.4** | 41.9 | 72.6 | 97.6 | 66.8 | 73.2 | 42.7 | 62.8 | 52.8 | 68.1 |
| TARA-Variant A (500 Iters) | 82.8 | 73.9 | 43.8 | 70.5 | 98.2 | **73.4** | 73.8 | 48.4 | 61.1 | 54.8 | 69.1 |
| TARA-Variant B (250 Iters) | **86.7** | 75.1 | 42.8 | **75.8** | 98.5 | 68.9 | 74.6 | 50.3 | **63.0** | 56.7 | 70.1 |
| TARA-Variant B (500 Iters) | 86.3 | 73.8 | 47.6 | 73.8 | **99.1** | 71.6 | **75.4** | **53.3** | 61.6 | **57.5** | **70.9** |

| Method | Cars | DTD | EuroSAT | GTSRB | MNIST | SUN397 | Avg Acc | RESISC45 | SVHN | Avg Acc | Avg Acc |
|---|---|---|---|---|---|---|---|---|---|---|---|
| | *Per-task accuracies of merged models, normalized to finetuned (%)* | | | | | | | | | | |
| TA | 82.3 | 75.1 | 53.0 | 40.7 | 52.4 | 96.7 | 66.7 | 68.6 | 34.7 | 51.7 | 63.0 |
| TIES | 81.9 | 74.8 | 73.7 | 33.2 | 63.4 | 95.6 | 70.4 | 68.9 | 31.6 | 50.3 | 65.4 |
| KnOTS-TIES | 82.5 | 72.9 | 53.8 | 47.8 | 61.7 | 94.9 | 69.0 | 67.5 | 35.1 | 51.3 | 64.5 |
| KnOTS-DARE-TIES | 82.7 | 72.2 | 53.1 | **48.5** | 61.5 | 94.3 | 68.7 | 66.7 | 35.4 | 51.0 | 64.3 |
| LoRA-LEGO | 81.1 | 72.3 | 55.6 | 42.8 | 62.6 | 94.7 | 68.2 | 65.2 | 33.9 | 49.5 | 63.5 |
| AdaMerging (250 Iters) | 82.7 | 75.4 | 72.6 | 37.6 | 70.6 | 97.2 | 72.7 | 69.1 | 38.5 | 53.8 | 67.9 |
| AdaMerging (500 Iters) | 79.6 | 74.2 | 72.5 | 36.5 | 60.0 | 97.9 | 70.1 | 69.2 | 41.1 | 55.2 | 66.4 |
| TARA-Variant A (250 Iters) | 84.9 | 75.7 | 76.3 | 40.5 | **77.4** | 97.5 | 75.4 | 69.5 | 42.1 | 55.8 | 70.5 |
| TARA-Variant A (500 Iters) | 82.5 | 75.6 | 75.7 | 41.1 | 71.9 | 98.3 | 74.2 | 69.4 | 45.9 | 57.7 | 70.1 |
| TARA-Variant B (250 Iters) | **86.7** | 77.0 | **79.5** | 42.8 | 76.3 | 98.2 | **76.8** | 70.5 | 45.6 | 58.1 | **72.1** |
| TARA-Variant B (500 Iters) | 86.2 | **77.3** | 78.8 | 44.7 | 70.0 | **98.6** | 75.9 | 70.4 | **47.7** | **59.1** | 71.7 |

merging baselines achieve limited performance with Hits@1 below 45%. KnOTS-TIES and KnOTS-DARE-TIES obtain small but consistent gains, while AdaMerging achieves further improvements when additional optimization iterations are allowed. TARA consistently outperforms all baselines. Both Variant A and Variant B benefit from longer training, with Variant A (500 Iters) reaching the highest Hits@1 (51.1%) and strong overall balance, while Variant B (500 Iters) delivers the best Hits@5 (85.1%). These results highlight that our merging strategy effectively integrates task-specific knowledge across datasets and yields more general models under the challenging joint-task setting. The full results are provided in Tab. 10.

**Two-Task Trade-Off Analysis.** We examine whether preference-aware merging can trace an accuracy trade-off. Figure 3 shows results on the CLIP ViT-B/32 backbone. For the GTSRB-SVHN pair, sweeping 30 preference values and merging the two adapters with TARA yields a smooth trade-off curve (line). Baseline mergers (AdaMerging, KnOTS, LoRA-LEGO, TA) appear as isolated operating points. By jointly accounting for *subspace coverage* (retaining effective rank) and *anisotropy* (reweighting directional emphasis), TARA produces merged models that reflect diverse preferences and in turn typically lie above the baseline points, offering stronger accuracy trade-offs at comparable performance levels and enabling selection of an operating point by adjusting the preference.

**Preference Sensitivity Analysis.** To assess robustness to changes in the preference vector, we adopt the following setting. We fix the GTSRB and SVHN preferences to 0.125 each and randomly sample the remaining task preferences on the simplex so that all entries sum to 1. We sample 30 preference vectors and merge eight CLIP ViT-B/32 adapters with TARA and with AdaMerging, then record accuracies on GTSRB and SVHN. In Fig. 4 each point corresponds to one sampled preference and its resulting two-task accuracies. For AdaMerging we use a weighted sum of per-task pseudo-objectives.

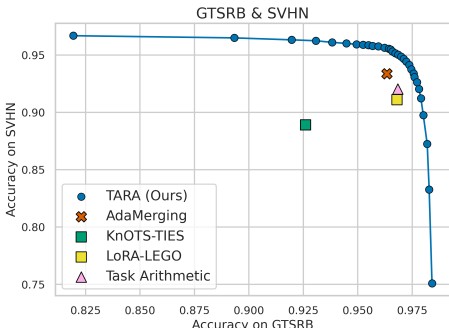

Figure 3: **Two-Task Merging Results.**

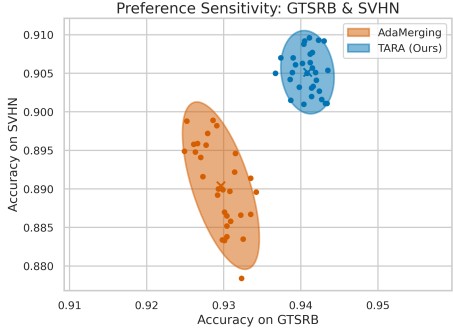

Figure 4: **Preference Sensitivity** of TARA and AdaMerging.

Compared to AdaMerging, TARA produces a tighter point cloud with lower empirical covariance and more stable two-task performance under preference perturbations, consistent with its direction-aware handling during LoRA merging.

**Evaluation of Generalization to Unseen Tasks.** We assess the generalization performance of TARA following the evaluation protocol of Yang et al. (2023). In this setting, *seen tasks* are those for which LoRA fine-tuned checkpoints are available and included during merging, whereas *unseen tasks* correspond to datasets without any associated LoRA checkpoints. As shown in Tab. 4, models are merged on six seen tasks and subsequently evaluated on both the seen tasks themselves and two unseen tasks. The results demonstrate that vanilla baselines and LoRA-aware methods struggle to achieve strong performance on unseen datasets. In contrast, TARA consistently outperforms all baselines across both evaluation groups. Variant B (500 Iters) achieves the highest overall accuracy of 70.9% in the first split, while Variant B (250 Iters) reaches 72.1% in the second split. These results show that TARA preserves in-domain performance and generalizes better to unseen benchmarks.

## 5 RELATED WORK

**Model Merging.** Model merging integrates knowledge from separately fine-tuned models without joint multi-task training, reducing computational cost while preserving task performance (Yang et al., 2024a). Core baselines include *Task Arithmetic*, which adds or subtracts task vectors (Ilharco et al., 2022), *RegMean*, which requires inner-product matrices of layer inputs for data-free fusion (Jin et al., 2022), *TIES*, which resolves sign conflicts and prunes small deltas (Yadav et al., 2023), and *DARE*, which sparsifies update deltas (Yu et al., 2024). *AdaMerging* learns layer-wise merge coefficients via entropy minimization to improve multi-task performance (Yang et al., 2023). For models trained on different data or from different initializations, permutation alignment preserves linear mode connectivity (Entezari et al., 2021; Ainsworth et al., 2022). *ZipIt* matches intermediate features to re-basin networks and align neurons across models, enabling training-free layer alignment (Stoica et al., 2023). Uncertainty-aware combinations mitigate mismatch through Fisher-weighted averaging (Matena & Raffel, 2022) and uncertainty or gradient matching (Daheim et al., 2023). Preference-aware settings motivate multi-objective formulations such as Pareto merging (Chen & Kwok, 2024) and smooth scalarization (Lin et al., 2024). Overall, recent work advances simple averaging into alignment- and uncertainty-aware schemes with explicit trade-off control.

**LoRA-aware Merging.** Low-Rank Adaptation (LoRA) (Hu et al., 2022) is a widely used fine-tuning scheme for large networks, including foundation models such as LLMs (Dubey et al., 2024). Conventional parameter-space merging methods (Jin et al., 2022; Ilharco et al., 2022; Yadav et al., 2023; Yu et al., 2024) transfer poorly to LoRA adapters, motivating LoRA-aware approaches (Stoica et al., 2025; Zhao et al., 2025; Tang et al., 2025). In particular, Stoica et al. (2025) argue that LoRA-updated models exhibit weaker cross-model representation alignment than full-rank finetunes, and propose KNOTS, which concatenates adapter updates and applies an SVD to align them in a shared subspace before merging principal components. Zhao et al. (2025) introduce *minimal semantic units* (MSUs) and perform clustering to assemble a merged adapter with an adjustable effective rank. Related mixtures of LoRAs have also been explored for image generation, including multi-LoRA composition and concept mixing (Zhong et al., 2024; Gandikota et al., 2024; Zhuang et al., 2025). Additional related work and connections to our approach are discussed in Appendix B.

## 6 CONCLUSION

We revisited LoRA merging through two complementary lenses: *subspace coverage*, where LoRA updates collectively occupy a wide, low-redundancy subspace rather than collapsing onto a few shared directions, and *anisotropy*, the directional imbalance in how changes in the scale of LoRA rank-1 directions translate into loss changes. Ignoring either can collapse useful subspaces and bias loss-critical directions. Building on effective-rank and directional-sensitivity analyses, we introduce TARA-Merging to align direction weights with task preferences while preserving task-relevant subspaces. This maintains high subspace coverage and accounts for anisotropy via direction-wise reweighting, yielding consistent average gains over vanilla and LoRA-aware baselines across settings including per-task evaluation, joint evaluation, and transfer to unseen tasks. Addressing both subspace coverage and anisotropy is key to robust, general-purpose LoRA merging.

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

## A PROOF OF PROPOSITION 1

**Proposition 1** (Anisotropy Bounds). *Let $\sigma_{\max}(\boldsymbol{J})$ and $\sigma_{\min}(\boldsymbol{J})$ denote the largest and smallest singular values of $\boldsymbol{J}$. Then, for any coefficient vector $\phi$,*

$$\sigma_{\min}(\boldsymbol{J}) \, \|\phi\|_2 \ \leq \ \|\boldsymbol{J}\phi\|_2 \ = \ \|\Delta\boldsymbol{f}\|_2 \ \leq \ \sigma_{\max}(\boldsymbol{J}) \, \|\phi\|_2. \tag{4}$$

*When $\kappa(\boldsymbol{J}) = \sigma_{\max}(\boldsymbol{J})/\sigma_{\min}(\boldsymbol{J})$ is large, the map $\phi \mapsto \Delta\boldsymbol{f}$ is* anisotropic, *and equal-norm LoRA-direction updates need not yield proportional task-loss changes.*

*Proof.* Let $\boldsymbol{J} \in \mathbb{R}^{N \times K}$ be the restricted task-loss Jacobian with entries $\boldsymbol{J}_{i,k} = \langle \nabla f_i(\boldsymbol{W}), \, \boldsymbol{S}_k \rangle$, and take its SVD $\boldsymbol{J} = \boldsymbol{U}\boldsymbol{\Sigma}\boldsymbol{V}^\top$, where $\boldsymbol{U} \in \mathbb{R}^{N \times q}$ and $\boldsymbol{V} \in \mathbb{R}^{K \times q}$ have orthonormal columns ($\boldsymbol{U}^\top \boldsymbol{U} = \boldsymbol{V}^\top \boldsymbol{V} = \boldsymbol{I}_q$), $\boldsymbol{\Sigma} = \mathrm{diag}(\sigma_1, \ldots, \sigma_q)$ with $\sigma_1 \geq \cdots \geq \sigma_q > 0$, and $q = \mathrm{rank}(\boldsymbol{J})$. For any coefficient vector $\phi \in \mathbb{R}^K$, define $\tilde{\phi} = \boldsymbol{V}^\top \phi \in \mathbb{R}^q$. By orthogonal invariance of the Euclidean norm,

$$\|\boldsymbol{J}\phi\|_2 = \|\boldsymbol{U}\boldsymbol{\Sigma}\boldsymbol{V}^\top\phi\|_2 = \|\boldsymbol{\Sigma}\boldsymbol{V}^\top\phi\|_2 = \|\boldsymbol{\Sigma}\tilde{\phi}\|_2. \tag{11}$$

The second equality holds since multiplication by $\boldsymbol{U}$ preserves the $\ell_2$ norm: for any $y \in \mathbb{R}^q$, $\|\boldsymbol{U}y\|_2 = \|y\|_2$. Also, because $\boldsymbol{V}$ has orthonormal columns, $\|\boldsymbol{V}^\top\phi\|_2 \leq \|\phi\|_2$.

Since $\boldsymbol{\Sigma}$ is diagonal,

$$\|\boldsymbol{J}\phi\|_2^2 = \|\boldsymbol{\Sigma}\tilde{\phi}\|_2^2 = \sum_{i=1}^{q} \sigma_i^2 \, \tilde{\phi}_i^2. \tag{12}$$

For the upper bound, use $\sigma_i \leq \sigma_{\max}(\boldsymbol{J})$ and $\|\tilde{\phi}\|_2 \leq \|\phi\|_2$ to obtain

$$\|\boldsymbol{J}\phi\|_2^2 = \sum_{i=1}^{q} \sigma_i^2 \, \tilde{\phi}_i^2 \leq \sigma_{\max}(\boldsymbol{J})^2 \sum_{i=1}^{q} \tilde{\phi}_i^2 \leq \sigma_{\max}(\boldsymbol{J})^2 \|\phi\|_2^2, \tag{13}$$

hence $\|\boldsymbol{J}\phi\|_2 \leq \sigma_{\max}(\boldsymbol{J}) \, \|\phi\|_2$.

For the lower bound, decompose $\phi = \phi_{\mathrm{R}} + \phi_{\mathrm{N}}$ with $\phi_{\mathrm{R}} := \boldsymbol{V}\boldsymbol{V}^\top\phi \in \mathrm{range}(\boldsymbol{V})$ and $\phi_{\mathrm{N}} := (\boldsymbol{I} - \boldsymbol{V}\boldsymbol{V}^\top)\phi \in \ker(\boldsymbol{J})$. Equivalently, $\tilde{\phi} = (\tilde{\phi}_1, \ldots, \tilde{\phi}_q)$ are the coordinates of $\phi_{\mathrm{R}}$ in the $\boldsymbol{V}$-basis. Then

$$\|\boldsymbol{J}\phi\|_2^2 = \sum_{i=1}^{q} \sigma_i^2 \, \tilde{\phi}_i^2 \geq \sigma_{\min}(\boldsymbol{J})^2 \sum_{i=1}^{q} \tilde{\phi}_i^2 = \sigma_{\min}(\boldsymbol{J})^2 \|\phi_{\mathrm{R}}\|_2^2, \tag{14}$$

which implies $\|\boldsymbol{J}\phi\|_2 \geq \sigma_{\min}(\boldsymbol{J}) \, \|\phi_{\mathrm{R}}\|_2$. If one restricts to the admissible subspace that excludes the nullspace (e.g., the LoRA span intersected with $\ker(\boldsymbol{J})^\perp$), then $\phi_{\mathrm{N}} = 0$ and $\|\phi_{\mathrm{R}}\|_2 = \|\phi\|_2$, yielding

$$\sigma_{\min}(\boldsymbol{J}) \, \|\phi\|_2 \ \leq \ \|\boldsymbol{J}\phi\|_2 \ \leq \ \sigma_{\max}(\boldsymbol{J}) \, \|\phi\|_2.$$

(Equivalently, interpret $\sigma_{\min}(\boldsymbol{J})$ as the smallest *nonzero* singular value on the feasible subspace.) $\square$

## B ADDITIONAL RELATED WORK

**Pre-Merging and Additional Model Merging.** Pre-merging studies clarify and formalize the linear-composition behavior that *Task Arithmetic* relies on: they analyze when parameter-space combinations approximate joint training via partial or tangent linearization (Liu et al., 2023; Tang et al., 2023; Jin et al., 2024; Ortiz-Jimenez et al., 2024) and connect these to NTK-style local linearization (Jacot et al., 2018). Building on this foundation, recent formulations explore optimal-transport fusion for Transformers (Imfeld et al., 2023), cycle-consistent multi-model merging (Crisostomi et al., 2024), deriving layer-wise coefficients from model-internal statistics (Huang et al., 2024b), permutation with least-squares alignment (Nasery et al., 2025), constructing merging recipes from collections of fine-tuned models with diverse hyperparameters (Wortsman et al., 2022a) with early evidence on CLIP that averaging robustness-oriented fine-tunes can improve zero-shot robustness (Wortsman et al., 2022b), evolutionary search for recipe discovery (Akiba et al., 2025), sparsity-

and magnitude-aware sampling to reduce interference (Deep et al., 2024), and dynamic Fisher weighting guided by Bayesian optimization (Lee et al., 2025). AIM (Nobari et al., 2025) uses the information from the activation space of LLMs for merging. Within *LoRA-targeted* methods, a training-free framework decouples direction and scale and orthogonalizes adapter directions before merging (Zheng et al., 2025). Zhang et al. (2023) compose PEFT modules via weight-space arithmetic, enabling flexible transfer without extra training. Zhang & Zhou (2025) enforce orthogonal LoRA subspaces pre-finetuning, reducing interference and preserving merge performance. Liu et al. (2025) adjusts task and layer-wise merging coefficients using activation/gradient sensitivity and cross-task transferability.

**LoRA Composition.** Dynamic routing or mixture-of-experts compositions learn to combine multiple LoRAs at inference time (Wu et al., 2024; Liao et al., 2025; Li et al., 2024; Tang et al., 2024), and few-shot dynamic composition improves cross-task transfer (Huang et al., 2024a; Horoi et al., 2025). In vision, multi-LoRA merging for multi-task recognition demonstrates modularity (Kesim & Helli, 2024). In diffusion, multi-LoRA composition and subject-style concept mixing further highlight modularity (Zhong et al., 2024; Gandikota et al., 2024; Zhuang et al., 2025; Zou et al., 2025; Shah et al., 2024; Yang et al., 2024b). These methods dynamically assign or route adapters per input or task in an MoE-like fashion, which alters the inference-time architecture and falls outside our scope.

## C  EXPERIMENTAL SETTINGS

### C.1  DATASETS

#### C.1.1  VISION BENCHMARKS

Following Stoica et al. (2025), we adopt a ViT-B/32 backbone pre-trained on ImageNet-21k Deng et al. (2009), fine-tuned with LoRA rank 16. Evaluation is conducted on eight standard image-classification datasets. Brief summaries are provided below.

**SUN397 (Xiao et al., 2010).** Large-scale scene recognition dataset with 397 categories and 108,754 images, each class having at least 100 samples.

**Cars (Krause et al., 2013).** Fine-grained car recognition covering 196 categories with 16,185 images, evenly split into train and test.

**RESISC45 (Cheng et al., 2017).** Remote sensing benchmark with 45 scene classes and 31,500 images, about 700 per class.

**EuroSAT (Helber et al., 2019).** Satellite image classification dataset of 27,000 images in 10 land-use categories across diverse regions.

**SVHN (Yuval, 2011).** Street View House Numbers dataset with 10 digit classes, 73,257 train images, 26,032 test images, and 500k+ extra samples.

**GTSRB (Stallkamp et al., 2011).** Traffic sign recognition dataset with 43 categories and over 50,000 labeled images.

**MNIST (LeCun, 1998).** Classic handwritten digit dataset of 70,000 grayscale images evenly distributed across 10 classes (60k train / 10k test).

**DTD (Cimpoi et al., 2014).** Texture classification dataset with 47 attributes and 5,640 images, about 120 per class.

#### C.1.2  LANGUAGE BENCHMARKS

In addition to the main benchmarks used in the paper, we evaluate our method on six classification datasets to assess its applicability to large language models. These datasets are used in conjunction with the LLaMA-3 8B model. Below, we briefly summarize each dataset.

**QNLI (Wang et al., 2018).** A question-answering dataset reformulated as sentence-pair classification. Each example is a (question, sentence) pair, and the label indicates whether the sentence contains the answer to the question (binary).

**MNLI (Williams et al., 2018).** A large, crowdsourced collection of sentence pairs annotated for natural language inference. Given a premise and a hypothesis, the task is to predict one of three labels: *entailment*, *contradiction*, or *neutral* (three-class).

**SNLI (Bowman et al., 2015).** A corpus of ∼570k human-written sentence pairs labeled for *entailment*, *contradiction*, or *neutral*, supporting the standard NLI task (three-class).

**RTE (Dagan et al., 2005; Haim et al., 2006; Giampiccolo et al., 2007; Bentivogli et al., 2009).** A suite of textual entailment datasets compiled from the RTE1, RTE2, RTE3, and RTE5 challenges. Examples are drawn from news and Wikipedia. In the GLUE formulation, all RTE sets are converted to binary classification by collapsing *neutral* and *contradiction* into *not-entailment* for consistency.

**SICK (Marelli et al., 2014).** A dataset of around 10k sentence pairs built from image and video captions. Each pair is labeled for semantic relatedness (1-5 scale) and for textual entailment with three classes: *entailment*, *contradiction*, and *neutral*. It is widely used for evaluating compositional semantics, entailment, and similarity in a unified setting.

**SciTail (Khot et al., 2018).** An entailment dataset derived from multiple-choice science exams and web sentences. Each (premise, hypothesis) pair is labeled as *entails* or *neutral* (binary). The dataset contains 27,026 examples (10,101 *entails* and 16,925 *neutral*).

## C.2 TRAINING DETAILS

For the vision tasks, we fine-tuned CLIP (Radford et al., 2021) with a ViT-B/32 backbone. All models were optimized using AdamW (Loshchilov & Hutter, 2019) with a learning rate of 3e-4 and a cosine learning rate scheduler (Loshchilov & Hutter, 2017) with a weight decay of 1e-4.

For sequence classification, we fine-tuned LLaMA-3 (Dubey et al., 2024) with 8B parameters, following the setup of (Stoica et al., 2025). Optimization used AdamW with a learning rate of 3e-5 and a linear scheduler. A linear classification head with three outputs was added for natural language inference (entailment, contradiction, neutral), and in binary settings predictions for the unused class were disregarded.

To enable efficient adaptation, we applied LoRA (Hu et al., 2022) to both CLIP and LLaMA. Specifically, low-rank adapters were attached to the query, key, value, and output projection matrices of each transformer block. Unless otherwise stated, the default LoRA hyperparameters were: rank = 16, $\alpha = 16$, and dropout = 0.1.

## C.3 IMPLEMENTATION DETAILS

In this paper, we compare our approach against ten baselines, following the experimental protocols reported in their original works. For TA (Ilharco et al., 2022), we tune the scaling coefficient of task vectors across the range of $[0.1, 0.2, \ldots 1.0]$. For TIES (Yadav et al., 2023) and DARE-TIES (Yu et al., 2024), both the scaling coefficient $\lambda$ and pruning coefficient $\eta$ were selected via grid search. Specifically, $\lambda$ was varied over $[0.8, 0.9, \ldots, 1.8]$, while $\eta$ was explored over $[0.1, 0.2, \ldots, 0.9]$. For AdaMerging (Yang et al., 2023), all coefficients $\{\lambda_k^l\}_{n=1,l=1}^{N,L}$ (where $N$ is the number of tasks and $L$ the number of layers) are initialized to 0.3 before optimization with the entropy surrogate. For KnOTS (Stoica et al., 2025), we evaluate two variants: KnOTS-TIES and KnOTS-DARE-TIES, which combine KnOTS with TIES (Yadav et al., 2023) and DARE (Yu et al., 2024), respectively. For LoRA-LEGO (Zhao et al., 2025), we reimplement the algorithm following the original paper. We apply $k$-means clustering to LoRA Minimal Semantic Units (MSUs), where each MSU $u = [a, b]$ consists of a row $a$ from the LoRA down-projection matrix $A$ and the corresponding column $b$ from the up-projection matrix $B$. We experiment with cluster sizes $k \in \{8, 16, 32, 64, 128\}$, and reproduce the dual scaling strategies: parameter reweighting and output reweighting. Parameter reweighting rescales each cluster centroid $\mu$ to match the average norm of its members, $\mu' = \frac{\frac{1}{p}\sum_{i=1}^{p} \|s_i\|}{\|\mu\|}\mu$, compensating for the reduced norm after merging, while output reweighting scales the merged LoRA by $\frac{\sqrt{r}}{\sqrt{k}}$ (with $r$ the original LoRA rank and $k$ the merged rank) to stabilize variance. We evaluate every $k$ and both scaling strategies, reporting the best performance.

## C.4 EXPERIMENTAL SETTINGS OF PER-TASK AND JOINT-TASK EVALUATION

In addition to the standard per-task benchmark, we also adopt the "joint-task" evaluation introduced by Stoica et al. (2025). Unlike the per-task regime, where each dataset is treated independently, the joint-task aggregates the inputs and labels from all eight vision benchmarks into a single evaluation pool. After combining all label sets, duplicate classes are removed (for example, MNIST (Le-Cun, 1998) and SVHN (Yuval, 2011) share the same digit categories), resulting in 748 unique labels across the combined benchmark. This setup is particularly demanding because models must not only generalize within a task, but also discriminate among labels that appear across different datasets. In some cases, labels are semantically close or hierarchical, such as "islet" in SUN397 (Xiao et al., 2010) and "island" in RESISC45 (Cheng et al., 2017), which increases the difficulty of classification. To account for such ambiguity, evaluation is reported using Hits@$k$ metrics, where Hits@1 corresponds to top-1 accuracy and higher $k$ values allow partial credit when the correct label appears among the top-$k$ predictions. Joint-task protocol serves as an important benchmark for assessing the generality of merged models, as it directly tests whether a single model can operate across the diverse label space of multiple datasets.

## D SUPPLEMENTARY ANALYSIS: SUBSPACE COVERAGE

This appendix presents the exact formulations used in the main paper's subspace-coverage analysis. We form all stacks by vectorizing matrices and placing one vector per row. Let $\boldsymbol{W}_0 \in \mathbb{R}^{d \times m}$ be the pre-trained weight. For task $i \in \{1, \ldots, N\}$ with LoRA rank $r_i$, write $\Delta \boldsymbol{W}_i = \sum_{j=1}^{r_i} \boldsymbol{b}_{ij} \boldsymbol{a}_{ij}^\top$ with $\boldsymbol{b}_{ij} \in \mathbb{R}^d$ and $\boldsymbol{a}_{ij} \in \mathbb{R}^m$. Let $\mathrm{vec}(\cdot) : \mathbb{R}^{d \times m} \to \mathbb{R}^{dm}$ denote matrix vectorization. For any matrix $\boldsymbol{X}$, let $\{\sigma_k\}_{k=1}^{R_{\boldsymbol{X}}}$ be its nonzero singular values with $R_{\boldsymbol{X}} = \mathrm{rank}(\boldsymbol{X})$ and define

$$p_k = \frac{\sigma_k^2}{\sum_{j=1}^{R_{\boldsymbol{X}}} \sigma_j^2}, \qquad \mathrm{erank}(\boldsymbol{X}) = \exp\Big( -\sum_{k=1}^{R_{\boldsymbol{X}}} p_k \log p_k \Big).$$

The following entries are used in our analysis. For each entity, we define the stacked matrix and report its subspace coverage via effective rank.

- **Per-task sum** — compute coverage per task and then aggregate across tasks:

$$\boldsymbol{X}_i = \begin{bmatrix} \mathrm{vec}\big(\boldsymbol{b}_{i1} \boldsymbol{a}_{i1}^\top\big) \\ \vdots \\ \mathrm{vec}\big(\boldsymbol{b}_{ir_i} \boldsymbol{a}_{ir_i}^\top\big) \end{bmatrix} \in \mathbb{R}^{r_i \times dm}, \qquad \mathrm{PerTaskSum} = \sum_{i=1}^{N} \mathrm{erank}(\boldsymbol{X}_i).$$

- $\Delta \boldsymbol{W}$ **stack (LoRA-agnostic)** — coverage of task updates when the LoRA rank-1 structure is ignored. Stack the task updates as

$$\boldsymbol{X}_{\mathrm{agnostic}} = \begin{bmatrix} \mathrm{vec}\big(\Delta \boldsymbol{W}_1\big) \\ \vdots \\ \mathrm{vec}\big(\Delta \boldsymbol{W}_N\big) \end{bmatrix} \in \mathbb{R}^{N \times dm}, \qquad \mathrm{erank}(\boldsymbol{X}_{\mathrm{agnostic}}).$$

- **Rank-1 stack (LoRA-aware)** — coverage when rank-1 directions from all adapters are retained. Stack all rank-1 directions as

$$\boldsymbol{X}_{\mathrm{aware}} = \begin{bmatrix} \mathrm{vec}\big(\boldsymbol{b}_{11} \boldsymbol{a}_{11}^\top\big) \\ \vdots \\ \mathrm{vec}\big(\boldsymbol{b}_{Nr_N} \boldsymbol{a}_{Nr_N}^\top\big) \end{bmatrix} \in \mathbb{R}^{(\sum_i r_i) \times dm}, \qquad \mathrm{erank}(\boldsymbol{X}_{\mathrm{aware}}).$$

**Other modules.** Supplementary results in Fig. 5 show the same qualitative pattern across additional modules (e.g., Attention and MLP layers). The ordering *Per-task sum* $\geq$ *Rank-1 stack (LoRA-aware)* $\geq \Delta \boldsymbol{W}$ *stack (LoRA-agnostic)* consistently holds, and the LoRA-aware stack retains roughly $60\% \sim 70\%$ of the per-task sum. The gap between LoRA-aware and LoRA-agnostic reflects subspace collapse under interpolation-based merging.

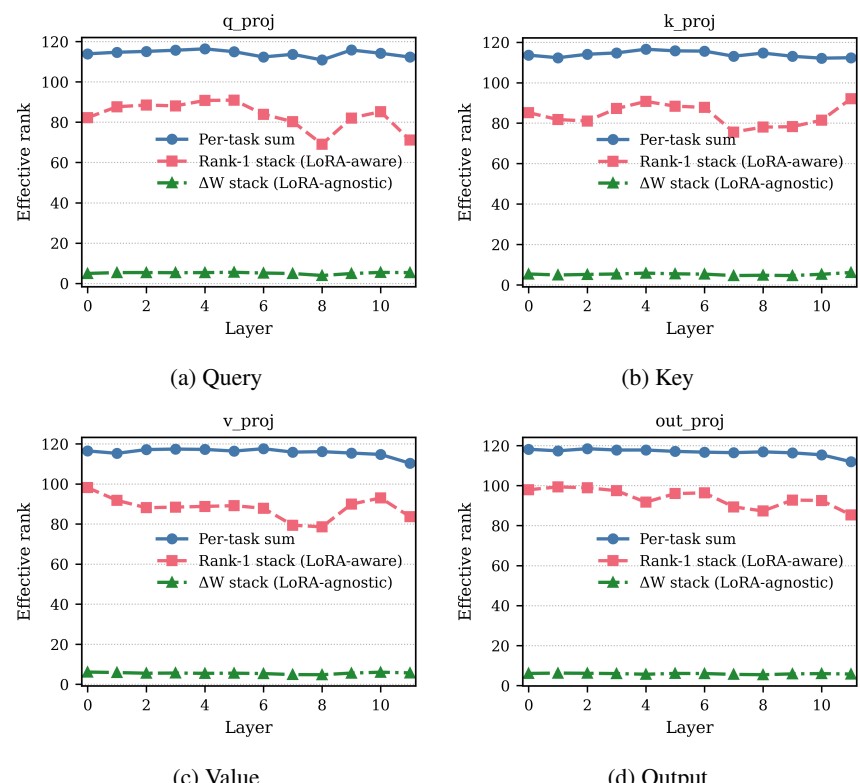

Figure 5: **Effective Rank** across layers for each module. Higher curves indicate broader subspace coverage. The gap between $\Delta W$ *stack* and *Rank-1 stack* reflects merge-induced collapse.

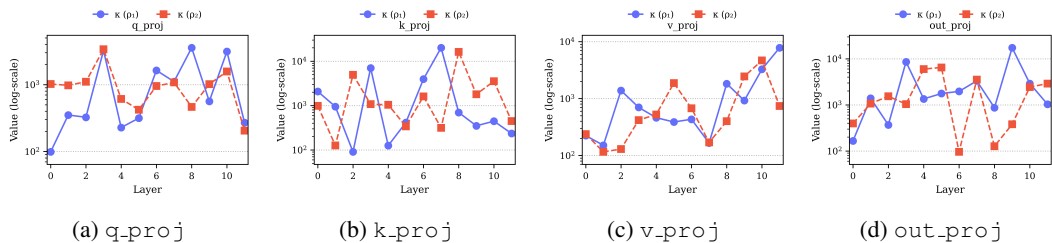

Figure 6: **Condition Number Anisotropy $\kappa$ (RAW Basis).** Layer-wise condition number $\kappa(\rho)$ per module under the non-orthogonal LoRA basis (RAW). Larger $\kappa$ indicates stronger *within-preference* directional concentration of loss sensitivity. (Here, $\rho_1$ uniformly weights all tasks, whereas $\rho_2$ assigns all weight to a single task (one-hot).)

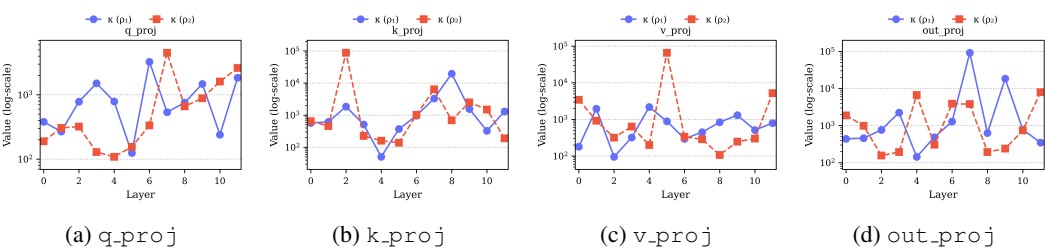

Figure 7: **Condition Number Anisotropy $\kappa$ (SVD Basis).** Layer-wise condition number $\kappa(\rho)$ per module under the orthogonal LoRA basis (SVD). Larger $\kappa$ indicates stronger *within-preference* directional concentration of loss sensitivity. (Here, $\rho_1$ uniformly weights all tasks, whereas $\rho_2$ assigns all weight to a single task (one-hot).)

# E SUPPLEMENTARY ANALYSIS: ANISOTROPY

**Within-Preference Anisotropy.** We measure how unevenly a single preference $\rho \in \Delta_{N-1}$ concentrates loss sensitivity onto LoRA directions. Let $g(\rho; W) = \sum_{i=1}^{N} \rho_i \nabla f_i(W)$ be the scalarized gradient at parameters $W$ obtained by *averaging* task LoRA updates and then scaling by 0.3 (as Task Arithmetic (Ilharco et al., 2022)). Given a basis of LoRA directions $\{S_k\}_{k=1}^{K}$ (either the RAW stack of rank-1 factors or the shared SVD-orthonormal basis), define the directional sensitivities with *condition-number*. Larger $\kappa$ indicates stronger *within-preference* concentration of sensitivity onto a few directions (greater anisotropy), while smaller $\kappa$ indicates a more balanced use of the LoRA subspace. We report layer-wise $\kappa(\rho; W)$ for each module (q_proj, k_proj, v_proj, out_proj) under both RAW and SVD bases in Fig. 6 and Fig. 7. Here, $\rho_1$ uniformly weights all tasks, whereas $\rho_2$ assigns all weight to a single task (one-hot).

Figure 8 compares, for each module and layer, the singular-value energy distribution of the task-direction sensitivity matrix under the RAW (rank-1 factor stack) and SVD (shared orthonormal) bases. Each curve shows the energy fraction $\sigma_k^2 / \sum_j \sigma_j^2$ versus component index $k$. A larger leading mass indicates stronger anisotropy with sensitivity concentrated in a few directions, while flatter curves indicate a more balanced use of the LoRA subspace.

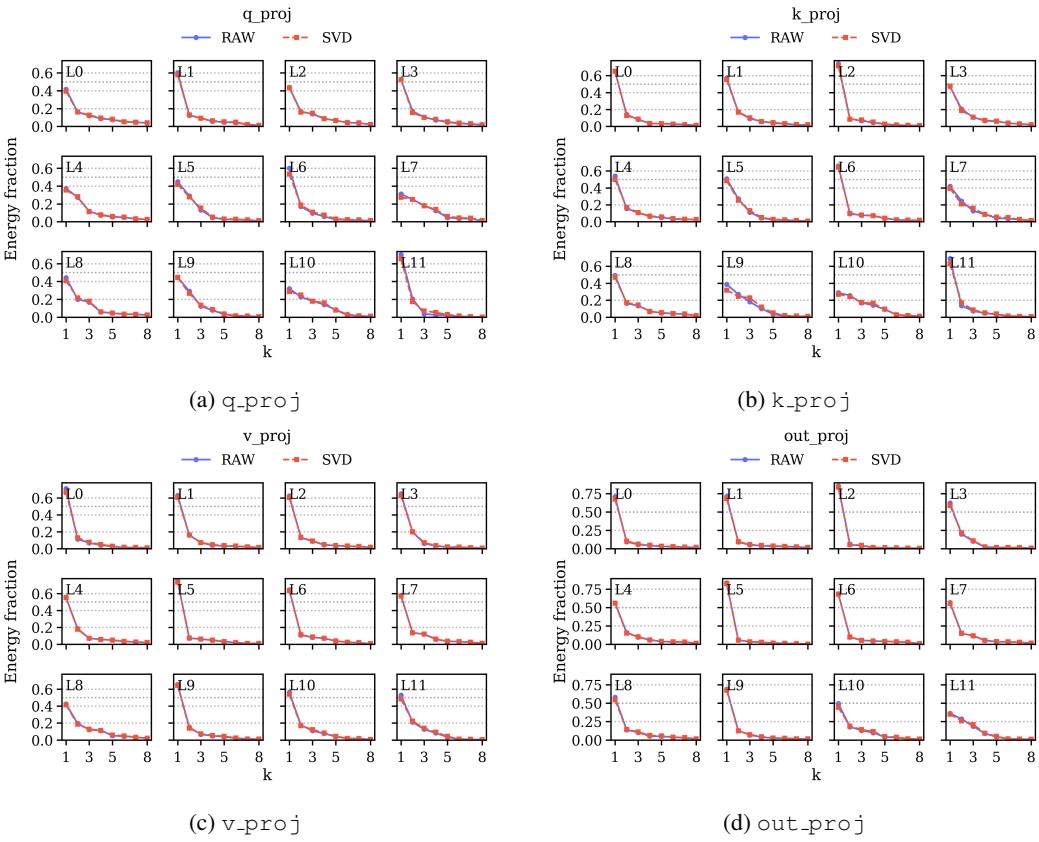

(a) q_proj

(b) k_proj

(c) v_proj

(d) out_proj

Figure 8: **Layer-wise Directional-Sensitivity Spectra (RAW vs. SVD).** For each module and layer, we form the task-direction sensitivity matrix with entries $J_{i,k} = \langle \nabla f_i(W), S_k \rangle$ and plot the scree curve of its singular values as energy fractions $\sigma_k^2 / \sum_j \sigma_j^2$ versus component index $k$. "RAW" stacks rank-1 LoRA factors directly, while "SVD" uses a shared orthonormal basis. The two spectra are overlaid in each panel. A larger leading mass indicates stronger concentration of sensitivity into a few modes, whereas flatter spectra indicate a more distributed use of LoRA directions.

**Directional-Sensitivity Misalignment in the SVD Basis.** Following Sec. 2.3, we conducted the same analysis with directional sensitivities projected onto the rank-1 LoRA directions obtained by SVD orthogonalization. We then computed a misalignment index $\xi(\boldsymbol{\rho}_1, \boldsymbol{\rho}_2) \in [0, 1]$ between the uniform and one-hot preferences across the LoRA layers. As shown in Figure 9, the resulting heatmap closely mirrors the raw-basis result in Fig. 2: substantial layer- and module-wise misalignment persists, indicating preference-dependent sensitive directions that remain even with SVD basis.

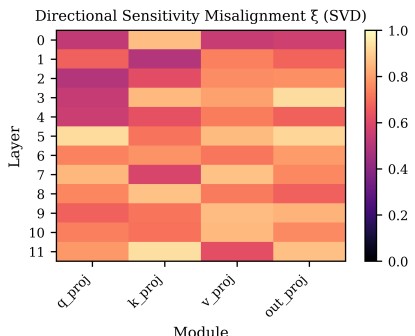

Figure 9: **Directional-sensitivity misalignment $\xi(\boldsymbol{\rho}_1, \boldsymbol{\rho}_2)$ in the SVD basis.**

## F  ADDITIONAL RESULTS

We complement the main-paper results (which use the KnOTS-released checkpoints (Stoica et al., 2025)) with additional experiments on our own LoRA checkpoints to gauge robustness across sources. Specifically, we fine-tune CLIP ViT-B/32 adapters with a learning rate different from that used by KnOTS, and then evaluate the same merging protocols. Table 5 reports per-task absolute accuracies for the fine-tuned adapters (upper panel) and, for each merger, taskwise accuracies normalized to the corresponding fine-tuned baseline (lower panel). Table 7 reports generalization to unseen tasks on our checkpoints. The results are consistent with the main paper, indicating that the advantages of preserving subspace coverage and addressing anisotropy hold across different fine-tuning learning rates. A comparison of checkpoints is provided in Tab. 6.

Table 5: Using our checkpoints. Per-task accuracy on eight image-classification benchmarks. We merge eight ViT-B/32 checkpoints, each finetuned with LoRA. The upper panel shows the per-task absolute accuracy of the finetuned baselines; the lower panel reports accuracy of merged models, normalized by their corresponding finetuned baseline (%).

| Method | Dataset | | | | | | | | |
|---|---|---|---|---|---|---|---|---|---|
| | Cars | DTD | EuroSAT | GTSRB | MNIST | RESISC45 | SUN397 | SVHN | Avg |
| | *Per-task absolute accuracies (%)* | | | | | | | | |
| Finetuned | 76.2 | 72.5 | 98.6 | 98.3 | 99.2 | 93.1 | 73.7 | 96.7 | 88.5 |
| | *Per-task accuracies of merged models, normalized to finetuned (%)* | | | | | | | | |
| ***Vanilla Merging-Gradient Free*** | | | | | | | | | |
| TA | 85.6 | 70.8 | 81.9 | 89.1 | 97.9 | 81.9 | 89.1 | 91.9 | 86.0 |
| TIES | 72.2 | 61.2 | 64.7 | 66.6 | 90.0 | 70.0 | 86.1 | 73.7 | 73.1 |
| DARE-TIES | 73.0 | 62.7 | 56.1 | 63.6 | 85.7 | 69.6 | 86.2 | 71.5 | 71.0 |
| ***Vanilla Merging-Gradient Based*** | | | | | | | | | |
| AdaMerging (250 Iters) | 87.1 | 71.2 | 94.6 | 94.0 | 98.5 | 85.9 | 87.0 | 92.0 | 88.8 |
| AdaMerging (500 Iters) | **87.7** | 71.2 | 96.3 | 94.4 | 98.7 | 87.3 | 86.8 | 91.9 | 89.3 |
| ***LoRA-aware Merging-Gradient Free*** | | | | | | | | | |
| SVD | 76.0 | 60.9 | 74.8 | 90.4 | 98.1 | 74.2 | 85.0 | 93.6 | 81.6 |
| Linear | 74.9 | 67.3 | 68.4 | 67.6 | 91.7 | 72.9 | 86.7 | 75.3 | 75.6 |
| KnOTS-TIES | 85.4 | 69.3 | 77.8 | 76.6 | 93.3 | 80.1 | 89.4 | 82.0 | 81.7 |
| KnOTS-DARE-TIES | 84.1 | 68.7 | 77.9 | 78.6 | 94.4 | 80.1 | **89.8** | 83.8 | 82.2 |
| LoRA-LEGO | 84.9 | 71.4 | 82.7 | 91.1 | 98.5 | 81.2 | 87.4 | 93.4 | 86.3 |
| ***LoRA-aware Merging-Gradient Based*** | | | | | | | | | |
| TARA-Variant A (250 Iters) | 86.5 | 72.0 | 94.4 | 95.0 | 98.8 | 85.3 | 87.4 | 93.5 | 89.1 |
| TARA-Variant A (500 Iters) | 87.2 | 72.5 | 96.5 | **95.7** | 99.0 | 87.0 | 88.0 | 93.7 | 89.9 |
| TARA-Variant B (250 Iters) | 86.1 | **72.7** | 95.7 | 95.6 | 99.0 | 86.4 | 88.0 | 94.2 | 89.7 |
| TARA-Variant B (500 Iters) | 85.8 | 72.0 | **96.8** | 96.4 | **99.1** | **88.0** | 87.4 | **95.2** | **90.1** |

Table 6: Comparison between merging methods between our checkpoints and KnOTS checkpoints.

| Method | Dataset | | | | | | | | |
|---|---|---|---|---|---|---|---|---|---|
| | Cars | DTD | EuroSAT | GTSRB | MNIST | RESISC45 | SUN397 | SVHN | Avg |
| | *Per-task absolute accuracies (%)* | | | | | | | | |
| Finetuned (KnOTS) | 74.0 | 58.3 | 99.0 | 92.7 | 99.3 | 88.4 | 64.5 | 96.2 | 84.1 |
| Finetuned (Ours) | 76.2 | 72.5 | 98.6 | 98.3 | 99.2 | 93.1 | 73.7 | 96.7 | 88.5 |
| | *Per-task accuracies of merged models, normalized to finetuned (%)* | | | | | | | | |
| TA (KnOTS) | 82.0 | 73.6 | 48.8 | 42.1 | 53.1 | 71.5 | 97.5 | 41.2 | 63.7 |
| TA (Ours) | 83.1 | 68.7 | 79.7 | 93.0 | 98.8 | 79.4 | 85.2 | 93.8 | 85.2 |
| TARA-Variant A (250 Iters, KnOTS) | 84.5 | 76.2 | 68.9 | 39.4 | 82.2 | 72.8 | 97.5 | 70.0 | 73.9 |
| TARA-Variant A (250 Iters, Ours) | 86.5 | 72.0 | 94.4 | 95.0 | 98.8 | 85.3 | 87.4 | 93.5 | 89.1 |
| TARA-Variant B (250 Iters, KnOTS) | 86.2 | 78.4 | 76.8 | 42.9 | 82.7 | 75.4 | 98.6 | 69.7 | 76.3 |
| TARA-Variant B (250 Iters, Ours) | 86.1 | 72.7 | 95.7 | 95.6 | 99.0 | 86.4 | 88.0 | 94.2 | 89.7 |

Table 7: Using our checkpoints. Generalization results on two unseen tasks when merging ViT-B/32 models trained on six tasks.

| Method | | Cars | DTD | GTSRB | RESISC45 | SUN397 | SVHN | Avg Acc | EuroSAT | MNIST | Avg Acc | Avg Acc |
|---|---|---|---|---|---|---|---|---|---|---|---|---|
| | | | | Seen Tasks | | | | | Unseen Tasks | | | All Tasks |
| | | | | | *Per-task accuracies of merged models, normalized to finetuned (%)* | | | | | | | |
| Task Arithmetic | | 85.6 | 73.0 | 94.7 | 82.4 | 88.4 | 95.0 | 86.5 | 47.3 | 78.1 | 62.7 | 80.6 |
| Ties-Merging | | 76.5 | 62.5 | 71.4 | 73.3 | 87.5 | 75.0 | 74.4 | 49.0 | 64.5 | 56.8 | 69.9 |
| KnOTS-TIES | | 85.0 | 69.4 | 83.9 | 81.1 | 89.4 | 84.8 | 82.3 | 56.6 | 73.4 | 65.0 | 78.0 |
| KnOTS-DARE-TIES | | 85.7 | 69.3 | 85.0 | 81.4 | **90.3** | 86.4 | 83.0 | 56.7 | 74.0 | 65.4 | 78.6 |
| LoRA-LEGO | | 85.2 | 72.9 | 94.9 | 82.3 | 88.3 | 94.8 | 86.4 | 46.9 | 77.8 | 62.4 | 80.4 |
| AdaMerging (250 Iters) | | 86.8 | 73.9 | 93.8 | 88.9 | 88.7 | 95.9 | 88.0 | 50.8 | 81.9 | 66.3 | 82.6 |
| AdaMerging (500 Iters) | | 87.1 | 73.8 | 93.4 | **91.2** | 88.4 | 96.2 | 88.4 | 52.2 | 83.1 | 67.7 | 83.2 |
| TARA-Variant A (250 Iters) | | 87.0 | 74.2 | 94.6 | 87.7 | 89.2 | 96.2 | 88.1 | 54.4 | 83.8 | 69.1 | 83.4 |
| TARA-Variant A (500 Iters) | | **87.4** | **74.9** | 94.9 | 89.6 | 89.2 | **96.7** | **88.8** | 57.4 | 86.6 | 72.0 | 84.6 |
| TARA-Variant B (250 Iters) | | 86.7 | 73.9 | 94.9 | 87.9 | 89.3 | 96.3 | 88.2 | 57.4 | 85.4 | 71.4 | 84.0 |
| TARA-Variant B (500 Iters) | | 87.2 | 74.7 | **95.4** | 89.4 | 89.3 | **96.7** | **88.8** | **63.4** | **87.7** | **75.5** | **85.5** |

| Method | | Cars | DTD | EuroSAT | GTSRB | MNIST | SUN397 | Avg Acc | RESISC45 | SVHN | Avg Acc | Avg Acc |
|---|---|---|---|---|---|---|---|---|---|---|---|---|
| | | | | | *Per-task accuracies of merged models, normalized to finetuned (%)* | | | | | | | |
| Task Arithmetic | | 87.2 | 75.6 | 86.4 | 95.6 | 98.4 | 90.6 | 89.0 | 60.4 | 61.6 | 61.0 | 82.0 |
| Ties-Merging | | 74.7 | 62.8 | 65.3 | 74.6 | 88.3 | 88.0 | 75.6 | 65.1 | 48.9 | 57.0 | 71.0 |
| KnOTS-TIES | | 86.7 | 70.8 | 80.8 | 83.3 | 94.1 | 90.4 | 84.3 | **67.5** | 57.2 | 62.4 | 78.8 |
| KnOTS-DARE-TIES | | 87.6 | 71.0 | 80.4 | 85.3 | 95.1 | 90.7 | 85.0 | 66.7 | 56.2 | 61.4 | 79.1 |
| LoRA-LEGO | | 87.0 | 75.3 | 86.0 | 95.9 | 98.5 | 90.2 | 88.8 | 60.0 | 61.5 | 60.8 | 81.8 |
| AdaMerging (250 Iters) | | 89.3 | 75.1 | 95.2 | 95.3 | 99.2 | 90.5 | 90.8 | 60.6 | 62.9 | 61.8 | 83.5 |
| AdaMerging (500 Iters) | | 89.7 | 74.7 | 95.0 | **99.4** | 94.4 | 91.0 | 91.0 | 60.1 | **63.7** | 61.9 | 83.7 |
| TARA-Variant A (250 Iters) | | 88.9 | **75.8** | 95.2 | 95.9 | 99.1 | 90.9 | 91.0 | 61.5 | 63.4 | **62.5** | 83.8 |
| TARA-Variant A (500 Iters) | | **89.9** | 75.0 | 97.1 | 96.2 | 99.3 | 90.9 | **91.4** | 61.2 | 63.5 | 62.4 | **84.1** |
| TARA-Variant B (250 Iters) | | 89.4 | 75.0 | 96.5 | 96.1 | 99.0 | 90.9 | 91.1 | 61.8 | 62.5 | 62.1 | 83.9 |
| TARA-Variant B (500 Iters) | | 89.7 | 74.9 | **97.4** | **96.4** | 99.2 | **91.0** | **91.4** | 60.7 | 58.6 | 59.7 | 83.5 |

**Merging Models with LoRA Rank 4.** LoRA is known to exhibit weaker cross-task alignment than full-rank fine-tuning, so LoRA-aware merging becomes especially important at low ranks. Using our CLIP ViT-B/32 checkpoints fine-tuned with rank-4 adapters, we evaluate vanilla and LoRA-aware merging in Tab. 8. Vanilla methods (e.g., TIES) degrade substantially, LoRA-aware baselines (KnOTS-TIES, LoRA-LEGO) recover more accuracy, and TARA achieves the strongest results overall: Variant B (500 iters) attains the best average normalized accuracy at **90.5**% and leads on most datasets. These rank-4 outcomes mirror the trends at higher ranks and underscore the benefit of modeling subspace coverage and anisotropy in low-rank LoRA merging.

Table 8: Using our checkpoints with LoRA (Rank-4). Per-task accuracy on eight image-classification benchmarks. We merge eight ViT-B/32 checkpoints, each finetuned with LoRA. The upper panel shows the per-task absolute accuracy of the finetuned baselines; the lower panel reports accuracy of merged models, normalized by their corresponding finetuned baseline (%).

| Method | Cars | DTD | EuroSAT | GTSRB | MNIST | RESISC45 | SUN397 | SVHN | Avg |
|---|---|---|---|---|---|---|---|---|---|
| | | | | | Dataset | | | | |
| | | | | | *Per-task absolute accuracies (%)* | | | | |
| Finetuned | 72.1 | 70.2 | 98.4 | 98.1 | 99.2 | 93.3 | 73.4 | 96.7 | 87.7 |
| | | | | *Per-task accuracies of merged models, normalized to finetuned (%)* | | | | | |
| ***Vanilla Merging-Gradient Free*** | | | | | | | | | |
| TA | 87.4 | 76.2 | 74.0 | 78.1 | 98.1 | 83.7 | 89.7 | 92.3 | 84.9 |
| TIES | 81.3 | 64.4 | 46.5 | 51.9 | 77.4 | 69.1 | 86.8 | 64.1 | 67.7 |
| DARE-TIES | 81.2 | 65.4 | 50.9 | 48.4 | 76.3 | 69.8 | 86.7 | 63.5 | 67.8 |
| ***Vanilla Merging-Gradient Based*** | | | | | | | | | |
| AdaMerging(250 Iters) | 87.6 | 75.4 | 89.5 | 89.5 | 98.0 | 86.2 | 89.9 | 90.8 | 88.4 |
| AdaMerging(500 Iters) | 87.9 | 75.1 | 94.7 | 92.0 | 98.2 | 86.9 | 89.5 | 91.1 | 89.4 |
| ***LoRA-aware Merging-Gradient Free*** | | | | | | | | | |
| SVD | 80.5 | 62.7 | 40.6 | 73.7 | 94.4 | 69.3 | 87.3 | 92.4 | 75.1 |
| Linear | 44.0 | 25.6 | 12.4 | 37.8 | 71.5 | 23.3 | 44.2 | 51.7 | 38.8 |
| KnOTS-TIES | 88.2 | 72.2 | 72.5 | 62.3 | 89.7 | 79.3 | 89.5 | 77.0 | 78.8 |
| KnOTS-DARE-TIES | 87.7 | 73.0 | 73.4 | 65.0 | 90.4 | 79.9 | 89.5 | 78.8 | 79.7 |
| LoRA-LEGO | 88.2 | 75.8 | 74.8 | 76.9 | 97.5 | 83.8 | **90.4** | 91.5 | 84.8 |
| ***LoRA-aware Merging-Gradient Based*** | | | | | | | | | |
| TARA-Variant A (250 Iters) | 88.9 | 75.7 | 89.9 | 89.3 | 98.2 | 85.9 | 90.2 | 91.6 | 88.7 |
| TARA-Variant A (500 Iters) | 88.7 | 76.4 | **94.9** | 92.1 | 98.3 | 86.8 | 90.0 | 92.0 | 89.9 |
| TARA-Variant B (250 Iters) | 89.4 | 76.4 | 89.9 | 90.3 | 98.3 | 86.7 | 90.5 | 92.5 | 89.2 |
| TARA-Variant B (500 Iters) | **89.8** | **76.7** | 94.6 | **92.8** | **98.4** | **88.0** | 90.1 | **93.2** | **90.5** |

**Time and Memory Cost.** Table 9 summarizes empirical efficiency in terms of *additional time* and peak VRAM. The upper panel reports gradient-free mergers, and the lower panel reports gradient-based mergers. For gradient-free methods, "additional time" denotes the cost of a single validation evaluation, with the number of trials used for linear hyperparameter search shown in parentheses. While each individual run is fast and memory-efficient, the full tuning process becomes expensive, with cost growing exponentially as more hyperparameters are introduced. For example, to tune the Task Arithmetic scaling factor $\lambda$ in $\boldsymbol{W}_{\text{merge}} = \boldsymbol{W}_0 + \lambda \sum_{i=1}^{N} \Delta \boldsymbol{W}_i$ via validation, a grid from $0.1$ to $1.0$ with step $0.1$ entails $10$ trials. If each evaluation takes $1.2$ minutes, the total comes to about $12$ minutes.

Gradient-based methods do not require such iterations, so their total time can be more favorable. TARA has comparable runtime to AdaMerging but yields better accuracy. Methods such as TARA-Variant B and KnOTS-TIES also incur a precompute SVD over all task vectors before multi-preference adaptation or linear hyperparameter search. For six LLaMA-3-8B models, TARA-Variant B spent 7.7 of its $16.4$ minutes on this SVD, leaving optimization time similar to Variant A and AdaMerging. Even for large models like LLaMA-3-8B, a merge finishes within ten minutes, underscoring the practical applicability of TARA.

In terms of memory, gradient-based methods use more VRAM than gradient-free methods because of backpropagation. However, the relative gap becomes smaller on large models such as LLaMA-3-8B, aligning with the practical goal of LoRA merging since LoRA is typically used to fine-tune large models. All times are reported in minutes (min) and memory as VRAM in MiB.

Table 9: Time and Memory Cost analysis. We apply rank-16 LoRA to the `q`, `k`, `v`, and output projections in every Transformer block. We merge eight CLIP models on an NVIDIA TITAN RTX and six LLaMA models on an NVIDIA GeForce RTX 3090. Baselines are grouped into gradient-free and gradient-based methods. For gradient-free methods, which tune a scale factor via validation, we report time per iteration and the number of iterations used for linear search (shown in parentheses), with all times measured in minutes (min). For gradient-based methods, we report total training time in minutes. Memory usage is reported as VRAM in MiB. Detailed explanation about the hyperparameters are at Appendix C.3.

| | Eight CLIP/ViT-B-32 | | Six LLaMA-3-8B | |
|---|---|---|---|---|
| Method | Additional Time (min) | VRAM (MiB) | Additional Timet (min) | VRAM (MiB) |
| ***Gradient Free*** | | | | |
| TA | 1.2 (10) | 674 | 5.6 (10) | 14848 |
| TIES | 1.2 (99) | 674 | 5.7 (99) | 14848 |
| KnOTS-TIES | 2.8 (99) | 2262 | 13.4 (10) | 17648 |
| LoRA-LEGO | 4.0 (20) | 1416 | 30.0 (20) | 17648 |
| ***Gradient Based*** | | | | |
| AdaMerging (500 Iters) | 4.1 | 4344 | 9.0 | 19918 |
| TARA-Variant A (500 Iters) | 4.1 | 4344 | 9.6 | 19922 |
| TARA-Variant B (500 Iters) | 5.1 | 5922 | 16.4 | 20040 |

Table 10: **Eight CLIP/ViT-B-32 Models Joint-Task Results.**

| Method | Metric | Joint-Task Performances (%) | | | | | | | | |
| | | Cars | DTD | EuroSAT | GTSRB | MNIST | RESISC45 | SUN397 | SVHN | Avg |
|---|---|---|---|---|---|---|---|---|---|---|
| Ensemble | Hits@1 | 58.5 | 41.3 | 16.9 | 29.5 | 35.8 | 54.2 | 62.4 | 25.1 | 40.7 |
| | Hits@3 | 83.6 | 61.4 | 31.4 | 59.5 | 58.0 | 78.2 | 84.0 | 44.3 | 63.1 |
| | Hits@5 | 91.0 | 72.4 | 40.0 | 73.0 | 69.1 | 85.9 | 89.5 | 55.2 | 72.6 |
| TA | Hits@1 | 60.7 | 40.7 | 15.3 | 38.8 | 31.8 | 59.7 | 61.9 | 29.2 | 43.5 |
| | Hits@3 | 84.9 | 63.7 | 23.0 | 66.1 | 48.4 | 83.6 | 83.9 | 50.1 | 65.2 |
| | Hits@5 | 92.0 | 74.0 | 31.0 | 77.9 | 55.7 | 90.2 | 89.9 | 61.6 | 74.0 |
| TIES | Hits@1 | 60.4 | 39.7 | 13.0 | 35.0 | 33.4 | 58.6 | 61.3 | 32.9 | 43.6 |
| | Hits@3 | 84.9 | 61.9 | 21.9 | 63.3 | 48.2 | 82.8 | 83.7 | 53.6 | 65.3 |
| | Hits@5 | 92.1 | 72.4 | 29.0 | 75.3 | 54.0 | 89.4 | 89.9 | 64.1 | 73.9 |
| DARE-TIES | Hits@1 | 60.8 | 39.3 | 12.8 | 33.7 | 34.3 | 57.5 | 60.4 | 35.5 | 44.0 |
| | Hits@3 | 85.3 | 61.4 | 18.1 | 63.6 | 50.2 | 82.3 | 82.6 | 57.8 | 66.4 |
| | Hits@5 | 92.6 | 73.0 | 20.9 | 74.6 | 55.7 | 89.0 | 89.1 | 69.5 | 75.1 |
| AdaMerging (250 Iters) | Hits@1 | 61.0 | 40.7 | 16.3 | 34.7 | 47.1 | 60.5 | 62.4 | 51.2 | 46.7 |
| | Hits@3 | 85.1 | 62.2 | 41.3 | 60.9 | 60.4 | 83.8 | 84.6 | 73.1 | 68.9 |
| | Hits@5 | 92.2 | 72.8 | 60.0 | 74.5 | 65.7 | 90.5 | 90.4 | 81.4 | 78.5 |
| AdaMerging (500 Iters) | Hits@1 | 58.7 | 37.9 | 18.1 | 36.7 | 51.6 | 57.4 | 63.1 | 61.6 | 48.1 |
| | Hits@3 | 83.1 | 59.5 | 56.9 | 63.6 | 71.0 | 81.3 | 84.8 | 85.1 | 73.2 |
| | Hits@5 | 91.0 | 70.5 | 72.7 | 75.9 | 80.7 | 89.1 | 90.8 | 92.8 | 83.0 |
| KnOTS-TIES | Hits@1 | 61.7 | 40.5 | 16.2 | 44.2 | 39.1 | 59.0 | 60.6 | 36.7 | 46.8 |
| | Hits@3 | 85.8 | 63.8 | 22.3 | 69.0 | 52.6 | 83.9 | 82.8 | 58.4 | 68.1 |
| | Hits@5 | 92.6 | 74.5 | 31.1 | 79.8 | 58.4 | 90.4 | 89.1 | 68.5 | 76.3 |
| KnOTS-DARE-TIES | Hits@1 | 60.4 | 40.3 | 15.9 | 41.9 | 34.6 | 58.4 | 60.4 | 34.8 | 45.2 |
| | Hits@3 | 85.0 | 63.5 | 21.1 | 68.4 | 50.0 | 83.8 | 82.4 | 56.5 | 66.9 |
| | Hits@5 | 92.2 | 74.5 | 26.6 | 78.9 | 55.9 | 90.4 | 88.9 | 67.4 | 75.3 |
| LoRA-LEGO | Hits@1 | 60.3 | 39.9 | 15.9 | 41.6 | 29.8 | 57.6 | 60.1 | 30.2 | 43.1 |
| | Hits@3 | 84.5 | 61.8 | 19.5 | 68.8 | 47.8 | 82.8 | 82.3 | 51.2 | 65.0 |
| | Hits@5 | 91.8 | 74.0 | 21.1 | 78.3 | 56.9 | 89.7 | 88.8 | 63.5 | 73.9 |
| TARA-Variant A (250 Iters) | Hits@1 | 62.5 | 41.0 | 14.9 | 36.6 | 52.6 | 61.4 | 62.7 | 58.3 | 48.7 |
| | Hits@3 | 86.4 | 62.7 | 39.9 | 63.9 | 64.0 | 84.7 | 84.9 | 77.4 | 70.5 |
| | Hits@5 | 93.1 | 73.9 | 59.7 | 76.8 | 68.2 | 90.9 | 90.5 | 84.3 | 79.7 |
| TARA-Variant A (500 Iters) | Hits@1 | 60.8 | 39.5 | 15.2 | 39.9 | 65.2 | 59.0 | 63.2 | 65.7 | 51.1 |
| | Hits@3 | 84.8 | 60.8 | 58.0 | 67.0 | 80.7 | 82.6 | 85.2 | 87.8 | 75.9 |
| | Hits@5 | 92.2 | 71.8 | 74.9 | 78.2 | 86.4 | 89.9 | 91.0 | 95.0 | 84.9 |
| TARA-Variant B (250 Iters) | Hits@1 | 63.8 | 41.0 | 14.4 | 39.9 | 59.9 | 63.5 | 63.4 | 58.7 | 50.6 |
| | Hits@3 | 87.2 | 63.0 | 50.5 | 66.5 | 71.4 | 86.0 | 85.3 | 78.0 | 73.5 |
| | Hits@5 | 93.5 | 74.0 | 70.0 | 78.3 | 76.3 | 92.0 | 91.0 | 85.7 | 82.6 |
| TARA-Variant B (500 Iters) | Hits@1 | 63.2 | 40.1 | 12.1 | 41.9 | 64.6 | 62.1 | 63.6 | 46.4 | 49.3 |
| | Hits@3 | 87.0 | 61.3 | 56.3 | 67.0 | 83.5 | 84.7 | 85.5 | 74.2 | 74.9 |
| | Hits@5 | 93.4 | 72.5 | 76.9 | 77.2 | 88.9 | 90.9 | 91.3 | 89.9 | 85.1 |

**Two-Task Pareto Fronts across Pairs.** Figure 10 plots accuracy trade-offs for four task pairs on CLIP ViT-B/32. Sweeping the preference with TARA yields smooth Pareto fronts that are consistently dominant over AdaMerging, that is, TARA attains higher accuracy for the same preference across most of the preference range, especially in the balanced region where both tasks must be retained. Baseline mergers appear as isolated points and frequently lie below or off our front. We attribute this dominance to TARA 's LoRA-aware design: it preserves subspace coverage, retaining useful low-rank directions across tasks, while reweighting anisotropic directions to mitigate interference, keeping the merged model competitive near each task-optimal end without collapsing in the middle.

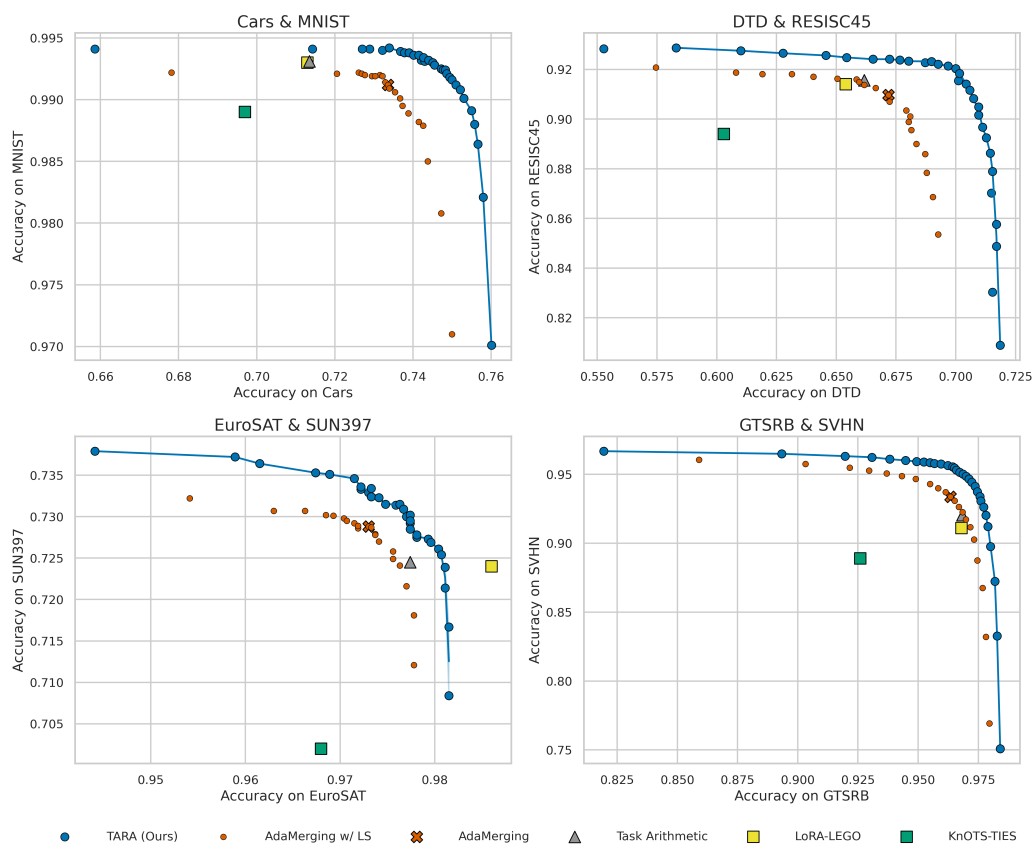

Figure 10: **Two-Task Trade-Offs on CLIP ViT-B/32.** Pairs shown from top-left to bottom-right: Cars-MNIST, DTD-RESISC45, EuroSAT-SUN397, GTSRB-SVHN.

**Robustness under Preference Perturbations.** Figure 11 tests robustness with 8 tasks by fixing the two focal-task weights to $0.125$ each (total $0.25$) and randomly choosing the other six weights so they are nonnegative and sum to $0.75$. Each scatter cloud shows 30 such samplings and the ellipses summarize their empirical covariance, with the mean marked by "×". Across all pairs, TARA produces tighter ellipses, indicating lower variance under preference perturbations, while AdaMerging shows larger spread. We also applied both a linear weighted-sum objective and a smooth Tchebycheff objective to TARA and to AdaMerging; the choice of objective yielded nearly identical ellipses and means.

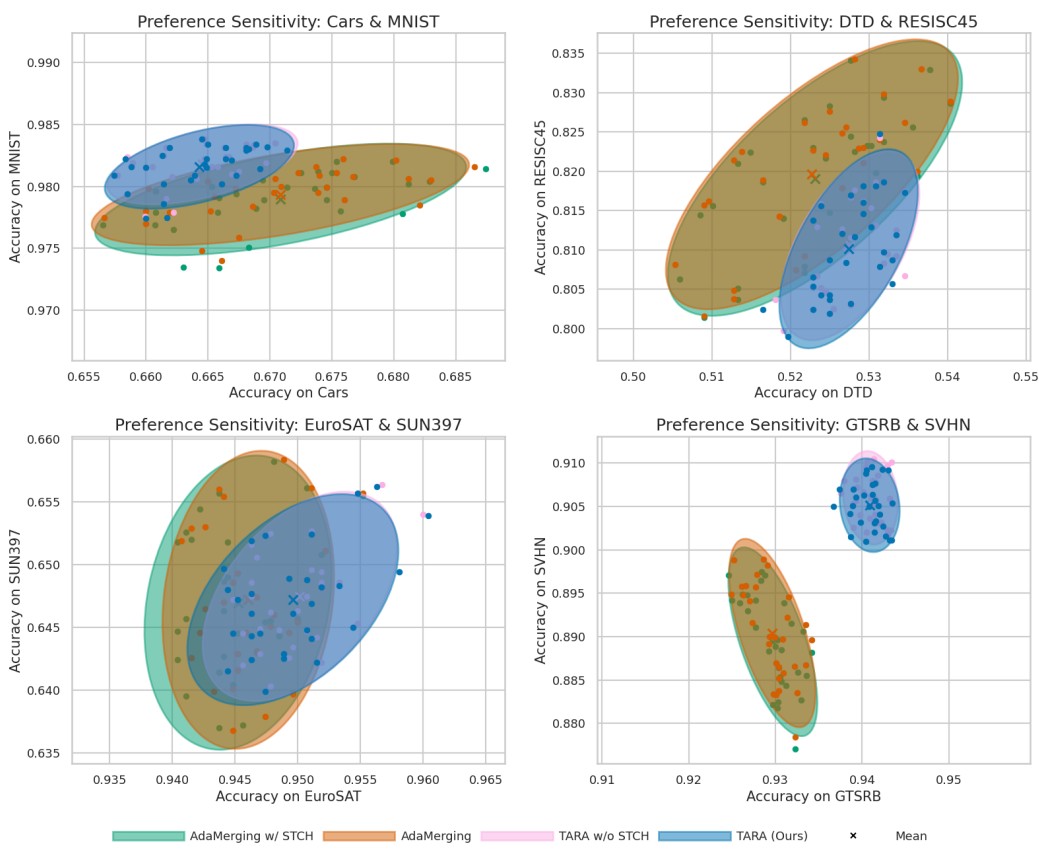

Figure 11: **Preference Sensitivity under Random Global Preferences.** Pairs shown from top-left to bottom-right: Cars-MNIST, DTD-RESISC45, EuroSAT-SUN397, GTSRB-SVHN.

