# OpenReview forum: "Aligning Task-Rank Preferences: Subspace Coverage and Anisotropy in LoRA Merging"
_ICLR.cc/2026/Conference — ICLR 2026 Conference Withdrawn Submission_

### Official Review · Reviewer_oyDH · 2025-10-18

**Soundness:** 2
**Presentation:** 1
**Contribution:** 2
**Rating:** 2
**Confidence:** 4

**Summary:**

This paper focuses on LoRA Merging task and reveals two problems: subspace coverage and anisotropy. Proposed  TARA-Merging incurpincorpoartes demonstrates the effectiveness of proposed method.

**Strengths:**

Author conducts various experiments.

**Weaknesses:**

First, the writing clarity is insufficient. In the Abstract (Lines 21–23), the repetitive use of "with" creates redundant and hard-to-follow phrasing, and the authors should simplify this expression for readability. Additionally, while the paper emphasizes "subspace coverage" and "anisotropy" as key challenges, these issues are not unique to LoRA merging—they also exist in general model merging scenarios, which weakens the novelty. Furthermore, the paper lacks a motivation figure to confirm the existence of these two challenges.

Second, the rationale for key metrics is underdeveloped. In Line 116, the authors adopt "erank" (effective rank) to quantify subspace coverage but provide no justification for this choice—e.g., why erank is superior to existing rank-based metrics like Diff-eRank [1]?  Moreover, Figure 2 is confusing due to insufficient explanation.

Third, the dynamic selection of the two TARA variants is unaddressed. The paper introduces two variants—"Per-rank LoRA direction selection" (Variant A) and "Shared singular-direction selection" (Variant B)—but provides no guidance on when to use which. The absence of such guidance limits the practical applicability of the proposed framework.
[1] Diff-eRank: A Novel Rank-Based Metric for Evaluating Large Language Models

**Questions:**

Refer to weakness.

---

### Official Review · Reviewer_neHD · 2025-10-24

**Soundness:** 3
**Presentation:** 2
**Contribution:** 3
**Rating:** 6
**Confidence:** 3

**Summary:**

The paper introduces TARA-Merging, a LoRA-aware method that operates directly on the low-rank factors. It learns *direction-selection weights* for each rank-1 direction, allowing it to amplify critical directions and suppress those that cause interference. It optimizes these weights using a preference-weighted pseudo-loss. Two variants are offered: (A) reweighting original factors, and (B) reweighting a shared SVD basis.

TARA consistently and significantly outperforms all baselines (including vanilla methods like Task Arithmetic and LoRA-aware methods like KnOTS) across diverse vision (CLIP) and language (LLaMA-3 8B) benchmarks, showing superior per-task, joint-task, and generalization performance.

**Strengths:**

1. The core strength is the clear, empirically-backed diagnosis of the problem, centered on the intuitive concepts of "subspace coverage" and "anisotropy.”
2. The TARA method is a direct and logical solution to the identified problems, using direction-wise weights to combat anisotropy while its LoRA-aware design preserves the subspace.
3. The method is comprehensively tested on both vision and language models, using multiple evaluation protocols (per-task, joint-task, unseen tasks) and strong baselines. The Pareto-front analysis (Fig. 3) effectively demonstrates its superior trade-offs.

**Weaknesses:**

1. As a gradient-based method, TARA incurs an optimization cost in time and VRAM. While this may be offset by the heavy tuning costs of "gradient-free" alternatives, it still presents a compute hurdle.

2. Missing sensitive experiments on $\alpha$.

**Questions:**

1. In Section 2.2, is that fair to compare the per-task sum erank (First, multiple matrix) with the single erank (Second and Third, single matrix)?
2. Given the method's reliance on an entropy-based pseudo-loss, is the final merge quality sensitive to the domain of the unlabeled data used during optimization?
3. Are the TARA sensitive to the initial direction-selection weights $\phi$?

---

### Official Review · Reviewer_fxms · 2025-10-29

**Soundness:** 2
**Presentation:** 2
**Contribution:** 2
**Rating:** 4
**Confidence:** 3

**Summary:**

The paper introduces TARA-Merging for merging multiple LoRA adapters into a single model while honoring task-level preferences.

The key ideas are:
 (i) preserve subspace coverage by operating at the level of rank-1 LoRA directions, measured via effective rank, and (ii) address anisotropy in how task losses respond to direction updates by reweighting directions according to preference-weighted gradients. Two variants are proposed. Experiments on 8 CLIP vision tasks and 6 NLI tasks with LLaMA-3-8B report gains over vanilla and LoRA-aware baselines

**Strengths:**

- The coverage/anisotropy lenses are useful for thinking about LoRA merging, and the effective-rank/condition-number diagnostics are informative.
- Reweighting rank-1 LoRA directions (Variant A) and SVD-shared bases (Variant B) are straightforward to implement.
- Breadth of benchmarks.

**Weaknesses:**

1) Most headline tables report accuracy normalized to the fine-tuned baseline (Tabs. 1–2). This hides absolute accuracy and favors tasks with weaker baselines (e.g., DTD’s fine-tuned 58.3% in Tab. 1), making cross-method comparisons opaque. Some entries even exceed 100% (e.g., RTE 102.6–103.4% in Tab. 2), suggesting merged models can surpass the fine-tuned baselines, but without absolute numbers it is hard to judge effect sizes. Please add absolute accuracies for all merged models and include calibration metrics (e.g., ECE/Brier) given the entropy-minimization objective.

2) Underspecified choice of $R$ (Variant B) and other key hyperparameters.
Variant B requires selecting the SVD truncation $R$ after concatenating task updates (Sec. 3, Eqs. (8)–(9)), but the paper does not state how $R$ is chosen per model/layer/task set, nor provide an ablation on $R$. Likewise, weights $\phi$ are "signed unless otherwise noted," yet there is no ablation on sign constraints, norm penalties, or clipping, which affect stability and interference. These omissions limit reproducibility and could explain gains.

3) Coverage metric is sensitive to construction choices and not tied to task performance.  The effective rank analysis stacks vectorized LoRA directions and reports that the LoRA-aware stack retains ~70% of the per-task sum. But this metric depends on how directions are rescaled/duplicated across tasks and on the particular vectorization, and it is not shown to correlate with accuracy across methods/settings. A control where directions are norm-normalized and tasks are balanced would help.

4) Joint-task “ensemble” baseline seems under-specified and possibly unfair.  In the joint-task protocol , labels from 8 datasets are pooled, then Hits@k is computed. The ensemble baseline is ill-defined: how are logits from task-specialized heads mapped and calibrated into a common label space? Without score calibration across datasets, ensembles are disadvantaged relative to a single merged head. Please specify calibration (e.g., temperature per task, Platt scaling) or compare to a calibrated ensemble.

5) Gradient-free methods are tuned by grid search while TARA/AdaMerging are trained . The counts (e.g., 99 trials for TIES) and the ranges differ across baselines, but it is unclear whether the total budget (time/evals) is matched; moreover, Variant B benefits from a pre-compute SVD that partially “peeks” at all adapters before optimization.

6) The scalarization is defined with anchors $z_i = \min_W f_i(W)$ (ideal-point) in Eq. (10), which is crucial for Tchebycheff’s Pareto properties. But in implementation the paper replaces $z_i$ by “the entropy loss obtained when only task $i$’s adapter is applied,” which is generally *not* $\min_W f_i$. This breaks the standard guarantee that the scalarization maps preferences $\rho$ to (approximate) Pareto-optimal points, and can seriously distort trade-offs.

7) Variant B claims "direction weights then adjust these orthonormal rank-1 components" after defining S_ik = u_k v_ki^T where {u_k} are orthonormal left singular vectors and v_ki are per-task partitions (Eqs. (8)-(9)). While {u_k} are orthonormal, the full set {S_ik}_i,k is not orthonormal in the Frobenius inner product because for fixed k, the inner product ⟨S_ik, S_i'k⟩ = v_ki^T v_ki' ≠ 0 in general.

**Questions:**

1. On which split(s) are (i) the predictive-entropy objective and (ii) the anchors computed? Are any test images ever used for entropy-based tuning?
2. How is R selected per layer/model? Any ablation on R and on per-task normalization before SVD?
3. Please report absolute accuracies (not only normalized) for Tabs. 1–2 and calibration metrics
4.  Beyond two-task pairs, can you show multi-task results for several fixed $\rho$ vectors (e.g., uniform, sparse, long-tail) and quantify variance under perturbations?
5. Can you provide a budget-matched comparison (same wall-clock or eval count) for grid-search methods vs. TARA/AdaMerging?
6. How are per-dataset logits mapped/calibrated to the pooled label space? Is temperature scaling performed per task before ensembling?

---

### Official Review · Reviewer_cNqq · 2025-11-01

**Soundness:** 2
**Presentation:** 2
**Contribution:** 2
**Rating:** 2
**Confidence:** 4

**Summary:**

The paper, “Aligning Task–Rank Preferences: Subspace Coverage and Anisotropy in LoRA Merging,” studies LoRA-adapter merging through the lenses of subspace coverage and anisotropy. It proposes TARA (Task-Rank Anisotropy Alignment), which injects task preferences via a preference-weighted loss that reweights directions at both the task and rank levels. Experiments on vision and language benchmarks show consistent gains over prior baselines.

**Strengths:**

(1) Framing merging through subspace coverage and anisotropy is both interesting and novel.

(2) The proposed adaptation appears lightweight and straightforward to implement.

**Weaknesses:**

(1) Limited evaluation scope. The current 8-task ViT-B/32 setup is narrow. The authors should consider evaluating the widely used 8/14/20-vision task suites from [1] to assess scalability with more tasks and larger backbones. Additionally, evaluating the method on generative tasks would test applicability beyond classification tasks.

(2) Missing SOTA baselines. Important methods such as TSV-M and Isotropic Merging are absent; both have shown strong performance over Adamerging without additional adaptation. Including them would clarify TARA’s relative gains.

(3) Generality beyond LoRA. While motivated by LoRA, the core idea might extend to fully fine-tuned models. Exploring this would help delineate the scope of the approach.

(4) Analysis. It would be insightful to visualise the final rank- and task-specific weights after TARA adaptation and examine whether any salient patterns emerge.

References

[1] Wang, Ke, et al. Localizing task information for improved model merging and compression. arXiv:2405.07813 (2024).

**Questions:**

(1) Memory footprint. Does TARA require loading all task models simultaneously to learn rank/task-specific weights? If so, how does the method scale in memory with large backbones or many tasks, and are there practical workarounds?

**Details Of Ethics Concerns:**

No concerns.

---

### Note · Authors · 2025-11-12

**Comment:**

We thank all reviewers for their constructive feedback. After careful consideration, we have decided to withdraw our paper.

**Withdrawal Confirmation:**

I have read and agree with the venue's withdrawal policy on behalf of myself and my co-authors.